# Protein sequence editing defines distinct and overlapping functions of SKN-1A/Nrf1 and SKN-1C/Nrf2

Briar Jochim, Irini Topalidou (ID), Nicolas Lehrbach (ID) *

Basic Sciences Division, Fred Hutchinson Cancer Center, Seattle, Washinton, United States of America

* nlerhbach@fredhutch.org

## Abstract

The Nrf/NFE2L family of transcription factors regulates redox balance, xenobiotic detoxification, metabolism, proteostasis, and aging. Nrf1/NFE2L1 is primarily responsible for stress-responsive upregulation of proteasome subunit genes and is essential for adaptation to proteotoxic stress. Nrf2/NFE2L2 is mainly involved in activating oxidative stress responses and promoting xenobiotic detoxification. Nrf1 and Nrf2 contain very similar DNA binding domains and can drive similar transcriptional responses. In *C. elegans*, a single gene, *skn-1*, encodes distinct protein isoforms, SKN-1A and SKN-1C, that function analogously to mammalian Nrf1 and Nrf2, respectively, and share an identical DNA binding domain. Thus, the extent to which SKN-1A/Nrf1 and SKN-1C/Nrf2 functions are distinct or overlapping has been unclear. Regulation of the proteasome by SKN-1A/Nrf1 requires post-translational conversion of N-glycosylated asparagine residues to aspartate by the PNG-1/NGLY1 peptide:N-glycanase, a process we term 'sequence editing'. Here, we reveal the consequences of sequence editing for the transcriptomic output of activated SKN-1A. We confirm that activation of proteasome subunit genes is strictly dependent on sequence editing. In addition, we find that sequence edited SKN-1A can also activate genes linked to redox homeostasis and xenobiotic detoxification that are also regulated by SKN-1C, but the extent of these genes' activation is antagonized by sequence editing. Using mutant alleles that selectively inactivate either SKN-1A or SKN-1C, we show that both isoforms promote optimal oxidative stress resistance, acting as effectors for distinct signaling pathways. These findings suggest that sequence editing governs SKN-1/Nrf functions by tuning the SKN-1A/Nrf1 regulated transcriptome.

## Author summary

To maintain normal function, cells must adapt to stress through stress-responsive regulation of gene expression. SKN-1A/Nrf1 and SKN-1C/Nrf2 are two closely related transcription factors that play crucial roles in this process. We

**Data availability statement:** Gene expression (RNAseq) data are available from the Gene Expression Omnibus (GEO) with the identifier GSE279082. All other data are included in the manuscript and Supporting Information Files.

**Funding:** This work was supported by the National Institutes of Health (NIH) National Institute of General Medical Sciences (grant R35GM142728 to NL). This research was supported by the Genomics & Bioinformatics Shared Resource, RRID:SCR_022606, of the Fred Hutch/University of Washington/Seattle Children's Cancer Consortium (P30 CA015704). The funders had no role in study design, data collection and analysis, decision to publish, or preparation of the manuscript.

**Competing interests:** The authors have declared that no competing interests exist.

previously showed that an unusual post-translational modification that we term 'sequence editing', is important for SKN-1A/Nrf1-mediated control of proteasome subunit gene expression during proteotoxic stress. In this study, we investigated how 'sequence editing' defines the distinct roles of SKN-1A/Nrf1 and SKN-1C/Nrf2. We found that although sequence editing of SKN-1A/Nrf1 promotes activation of proteasome subunit genes, it dampens activation of other genes that are involved in the response to oxidative stress, a process primarily regulated by SKN-1C/Nrf2. Interestingly, we find that both SKN-1A/Nrf1 and SKN-1C/Nrf2 are required for optimal resistance to oxidative stress. Together, these findings suggest that 'sequence editing' fine-tunes SKN-1A/Nrf1-dependent gene regulation to enable adaptation to different stress conditions.

## Introduction

Animal cells must precisely control gene expression to adapt to diverse environmental and physiological conditions. Misregulation of stress-responsive and homeostatic gene regulatory programs is a driver of various diseases, including cancer, metabolic disorders, inflammatory disease, neurodevelopmental conditions, and age-associated neurodegeneration [1–3]. Therefore, insights into the mechanisms that orchestrate stress response and homeostatic gene expression pathways provide a basis for disease mitigation or prevention. Two closely related NFLE2L/Nrf family Cap'n'Collar (CnC) basic leucine zipper (bZip) transcription factors, NFE2L1 (Nrf1) and NFE2L2 (Nrf2), impact many aspects of cellular and organismal function through stress-responsive and homeostatic control of gene expression [4,5]. Given these roles, Nrf1 and Nrf2 represent attractive targets for therapeutic manipulation across several disease contexts [6–10]. In the nematode *C. elegans*, Nrf functions depend on a single gene, *skn-1* [11]. The *skn-1* gene gives rise to two major protein isoforms (SKN-1A and SKN-1C) through differences in transcription start sites and mRNA splicing. SKN-1A and SKN-1C function analogously to mammalian Nrf1 and Nrf2, respectively. How the distinct and overlapping functions of SKN-1A/Nrf1 and SKN-1C/Nrf2 are defined to mediate coherent stress responses remains an outstanding question. Addressing this issue is crucial for realizing the potential of therapeutic Nrf modulation.

SKN-1A/Nrf1 is an important regulator of proteostasis that controls proteasome biogenesis [4,12]. SKN-1A/Nrf1 activity is governed by an elaborate and conserved post-translational processing pathway [13]. SKN-1A/Nrf1 possesses an N-terminal transmembrane domain that targets it to the endoplasmic reticulum [14–16]. Once in the ER, SKN-1A/Nrf1 becomes N-glycosylated at certain asparagine (Asn) residues [16–20]. Although the precise pattern of N-glycosylation is not known, genetic and biochemical analysis indicates that SKN-1A is N-glycosylated at one or more of four N-glycosylation motifs, whereas Nrf1 is N-glycosylated at several or all of nine N-glycosylation motifs [21–23]. N-glycosylated SKN-1A/Nrf1 is released from the ER by the ER-associated degradation (ERAD) machinery and is typically rapidly

degraded by cytosolic proteasomes. Defects in proteasome function or proteostasis, as well as pharmacological prote-asome inhibition, increase the fraction of SKN-1A/Nrf1 that escapes degradation. Stabilized SKN-1A/Nrf1 then translo-cates to the nucleus, where it activates transcription of proteasome subunit genes [18–20,24]. This transcriptional activity requires two processing steps that occur after release of the N-glycosylated protein from the ER. Firstly, SKN-1A/Nrf1 undergoes a single endoproteolytic cleavage mediated by the DDI-1/DDI2 aspartic protease [20,25]. This cleavage event removes the N-terminal transmembrane domain, potentially facilitating the efficient release of SKN-1A/Nrf1 from the ER membrane and/or eliminating domain(s) that may interfere with its function in the nucleus [20,21,25–28]. Secondly, SKN-1A/Nrf1 is deglycosylated by the PNG-1/NGLY1 peptide:N-glycanase, a cytosolic deglycosylation enzyme that removes N-linked glycans from ERAD substrate glycoproteins [20,29]. Deglycosylation by PNG-1/NGLY1 deamidates N-glycosylated Asn residues, converting them to aspartate (Asp) [30]. This post-translational amino acid conversion, which we term 'sequence editing' is critical for activation of proteasome subunit genes by SKN-1A/Nrf1 [21,22].

Nrf2 regulates xenobiotic detoxification and oxidative stress responses [5,31]. Several studies suggest that SKN-1C is the major functional counterpart of Nrf2 in *C. elegans* [21,32–34]. Unlike SKN-1A/Nrf1, SKN-1C/Nrf2 lacks an ER-targeting transmembrane domain and so is not trafficked through the ER. Instead, SKN-1C/Nrf2 is regulated by cytosolic ubiquitin ligases. In mammalian cells, the stress responsive activation of Nrf2 is primarily controlled by Keap1 [35]. Keap1 acts as the substrate binding subunit of a CUL3/RBX1 ubiquitin ligase complex that triggers Nrf2's ubiquitination and deg-radation [36]. Keap1 binding to Nrf2 is reduced under oxidative stress and by some reactive xenobiotic compounds, lead-ing to Nrf2 stabilization and activation of Nrf2-dependent stress responses [37,38]. Keap1 is not conserved in *C. elegans,* instead, oxidative-stress and xenobiotic detoxification responses are negatively regulated by WDR-23 [39,40]. WDR-23 is the substrate adaptor of a CUL4/DDB1 ubiquitin ligase complex that binds to SKN-1C and is thought to mediate its proteasomal degradation [39]. WDR-23 may also negatively regulate SKN-1A levels, although the cleavage by DDI-1 likely renders SKN-1A non-responsive to regulation by WDR-23 under most circumstances [21,41]. The human ortholog, WDR23/DCAF11 binds to and inhibits Nrf2, but not Nrf1, suggesting a conserved mechanism that controls SKN-1C and Nrf2-dependent stress responses [42–44].

SKN-1A/Nrf1 and SKN-1C/Nrf2 bind to the same DNA sequence element [31,45]. Consequently, they may regulate overlapping sets of target genes, potentially allowing them to perform overlapping or redundant functions [46–49]. Indeed, although Nrf2 plays a significant role in regulating oxidative stress responses [5,31], multiple studies in mice show that Nrf1 also regulates oxidative stress response genes [47,50,51]. Importantly, *nrf1^-/- nrf2^-/-* double knockout mice show massively increased accumulation of reactive oxygen species compared to either single mutant, suggesting that Nrf1 and Nrf2 are redundant regulators of redox balance *in vivo* [47]. In contrast, Nrf2 does not regulate proteasome subunit gene expression, which is exclusively regulated by sequence edited Nrf1 [24,50,51]. However, it remains unclear whether sequence editing is needed for Nrf1-dependent control of oxidative stress responses. Interestingly, mutant forms of Nrf1 that do not undergo sequence editing are unable to regulate the proteasome, but retain partial functionality, raising the possibility that distinct post-translational processing events govern different Nrf1 functions [22,52]. In *C. elegans*, the func-tions of *skn-1* in oxidative stress have largely been studied using mutants or RNAi conditions that simultaneously inacti-vate both SKN-1A and SKN-1C, so the potential for distinct or overlapping functions of each isoform, and the relevance of sequence editing, remain unclear.

Here, we show that different patterns of sequence editing alter the transcriptional consequences of SKN-1A activa-tion. We find that sequence editing promotes activation of proteasome subunit genes, while dampening activation of genes associated with oxidative stress and xenobiotic detoxification. Using isoform-specific alleles, we dissect the dis-tinct and overlapping functions of SKN-1A and SKN-1C, revealing that both isoforms are required for optimal oxidative stress defenses. Our data indicate that sequence editing is required for SKN-1A to promote oxidative stress resistance and support a model in which SKN-1A and SKN-1C are controlled by distinct signaling pathways to coordinate transcrip-tional control of redox homeostasis. These findings suggest that N-glycosylation-dependent sequence editing fine-tunes

SKN-1A/Nrf1 function to orchestrate animal stress responses, shedding light on the mechanisms governing the distinct and cooperative roles of the SKN-1A/Nrf1 and SKN-1C/Nrf2 pathways, with implications for their roles in disease pathogenesis and longevity.

## Results

### Sequence editing defines distinct transcriptional outputs of SKN-1A and SKN-1C

We generated transgenic strains that express an N-terminally truncated form of SKN-1 (hereafter SKN-1t). SKN-1t is equivalent to SKN-1A lacking the first 167 amino acids or SKN-1C lacking the first 90 amino acids (Fig 1A). This truncated protein bypasses the normal regulatory mechanisms that limit activity in the absence of stress and so constitutively

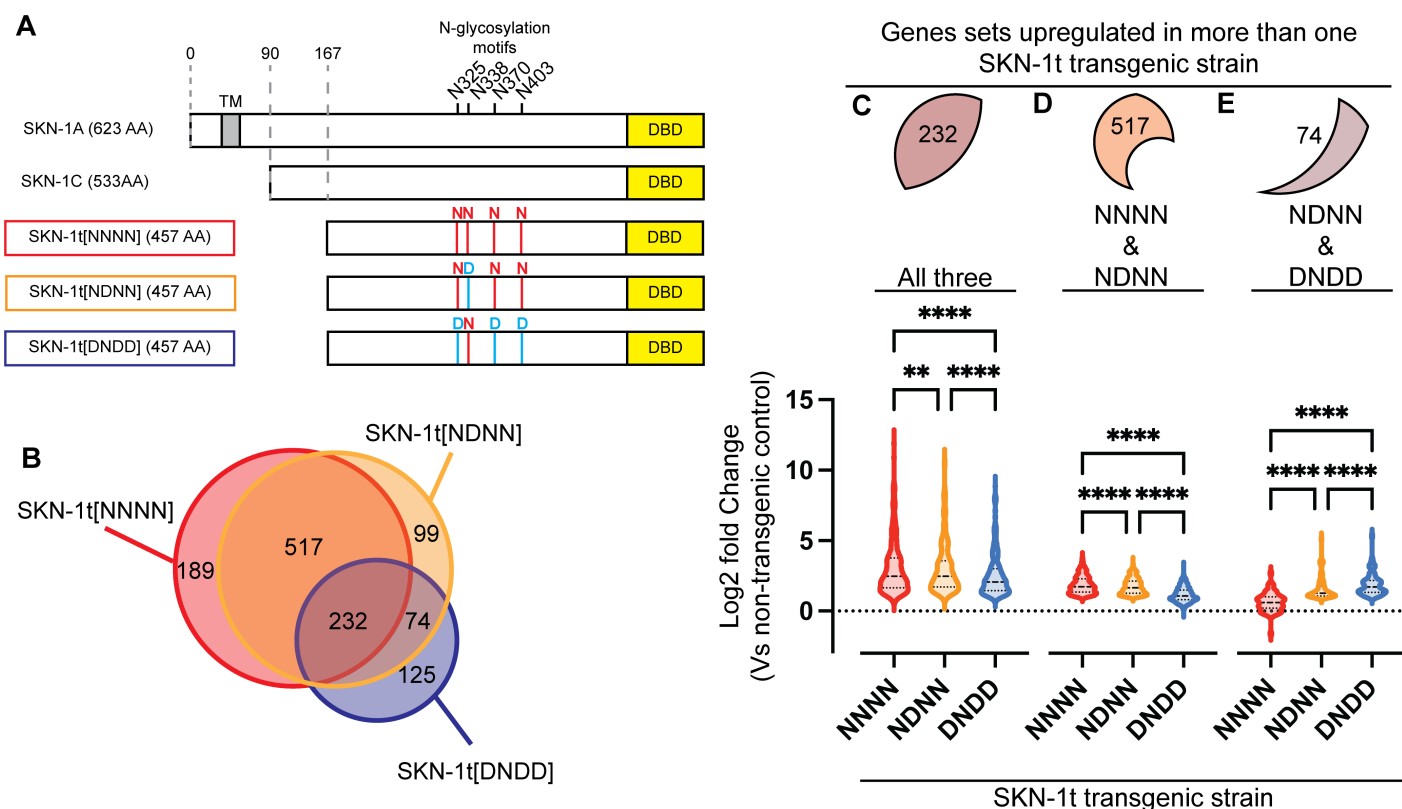

**Fig 1. Sequence editing alters the transcriptional output of activated SKN-1A.** A) Schematic showing the three forms of SKN-1t expressed by the transgenic animals subjected to RNAseq. The endogenous SKN-1A and SKN-1C proteins are shown for comparison. Each SKN-1t is expressed under the control of the ubiquitously active *rpl-28* promoter and with an N-terminal HA tag and a C-terminal GFP tag, which are not shown in the schematic. In SKN-1A, the locations of the four N-linked glycosylation motifs and the transmembrane domain (TM) are indicated. The position of the DNA binding domain (DBD) is shown for both SKN-1A and SKN-1C. B) Venn diagram showing the number of genes that are upregulated (>2-fold, FDR < 0.01), as identified by RNAseq of L4 stage SKN-1t transgenic animals and compared to wild type (non-transgenic) controls. C) Violin plot showing fold-upregulation of the 232 genes that are upregulated (>2-fold, FDR < 0.01) in all three SKN-1t transgenic strains. For each transgenic strain, the log2 Fold Change compared to the wild type (non-transgenic) control is plotted. These genes are upregulated to differing extents by the three transgenes. **** p < 0.0001, ** p < 0.01, repeated measures one-way ANOVA with Geisser-Greenhouse correction and Tukey's multiple comparisons test. D) Violin plot showing fold-upregulation of the 517 genes that are upregulated (>2-fold, FDR < 0.01) in SKN-1t[NNNN] and SKN-1t[NDNN] transgenic animals, but not in SKN-1t[DNDD] transgenics. These genes are more strongly upregulated by SKN-1t[NNNN] than SKN-1t[NDNN] and are skewed towards upregulation in SKN-1t[DNDD] transgenics. **** p < 0.0001, repeated measures one-way ANOVA with Geisser-Greenhouse correction and Tukey's multiple comparisons test. E) Violin plot showing fold-upregulation of the 74 genes that are upregulated (>2-fold, FDR < 0.01) in SKN-1t[NDNN] and SKN-1t[DNDD] transgenic animals, but not in SKN-1t[NDNN] transgenics. These genes are more strongly upregulated by SKN-1t[DNDD] than SKN-1t[NDNN]. **** p < 0.0001, repeated measures one-way ANOVA with Geisser-Greenhouse correction and Tukey's multiple comparisons test.

activates SKN-1 target genes [21]. SKN-1t lacks the ER-targeting transmembrane domain found at the N-terminus of SKN-1A, so it is not trafficked through the ER and does not undergo N-glycosylation-dependent sequence editing. We therefore engineered animals to express mutant forms of SKN-1t harboring Asn to Asp substitutions that mimic the effect of sequence editing and consequently alter SKN-1t function to model the effect of SKN-1A activation [21].

To investigate how sequence editing globally alters the transcriptional output of activated SKN-1A as compared to that of SKN-1C, we used RNAseq to compare animals expressing SKN-1t altered with different patterns of Asn to Asp substitutions (Fig 1A). To identify the set of genes upregulated by non-sequence-edited SKN-1C, we analyzed transcriptomes of animals expressing SKN-1t without any sequence changes at the four N-linked glycosylation sites (hereafter SKN-1t[NNNN]). The exact pattern(s) of N-glycosylation of SKN-1A are not known, so to identify genes upregulated by sequence-edited SKN-1A, we tested the effects of different patterns of sequence editing. First, we analyzed animals expressing SKN-1t with one Asn to Asp amino acid substitution (hereafter SKN-1t[NDNN]). This alteration mimics the effect of a single sequence editing event at N338 of SKN-1A and should therefore constitutively activate genes that would be activated if SKN-1A underwent limited N-linked glycosylation. Second, we analyzed animals expressing SKN-1t with three Asn to Asp amino acid substitutions (hereafter SKN-1t[DNDD]). We expect this construct to constitutively activate genes that would be activated if SKN-1A underwent extensive N-linked glycosylation (at N325, N370, and N403). We chose these two variants because both are competent to activate a proteasome subunit gene reporter and therefore could plausibly be involved in the SKN-1A-mediated response to proteasome inhibition [21], whilst allowing us to correlate global changes in gene expression with different extents of SKN-1A sequence editing. Transgenic animals were generated to express each form of SKN-1t from single copy transgenes with an N-terminal HA tag and C-terminal GFP tag. Each transgene is expressed at similar levels and transgenic strains show superficially normal development and fertility [21]. We did not include SKN-1t[N325D,N338D, N375D, N403D]-expressing animals in this analysis. Animals expressing this form of SKN-1t show a severely reduced growth rate and are therefore likely to show indirect changes in gene expression caused by this difference in developmental progression.

We identified differentially expressed genes in each transgenic strain compared to a wild type (non-transgenic) control (S1 Fig and S1 Table). Unsupervised principal component analysis indicates that all three transgenic strains cluster together with a distinct transcriptome from the wild type (S1 Fig). Because SKN-1 acts as a transcriptional activator, we focused our analysis on genes that are upregulated by >2-fold at a false-discovery rate (FDR) of <0.01 (Figs 1B and S1 and S1 Table). We identified 1235 genes that were upregulated in at least one of the SKN-1t transgenic strains. There is considerable overlap in gene upregulation between each strain (Fig 1B). Most (823/1235 = 67%) of the upregulated genes are increased in more than one of the strains analyzed, and many (232/1235 = 20%) are upregulated >2-fold in all three. This extensive overlap indicates that many SKN-1 target genes can be upregulated by both sequence-edited SKN-1A and non-edited SKN-1C. This suggests that distinct and overlapping function(s) of cytosolic and ER-associated SKN-1 isoforms could be achieved via regulation of partially overlapping transcriptional outputs.

### Sequence editing fine-tunes SKN-1A-dependent transcriptional programs

A substantial number of SKN-1t-upregulated genes are differentially activated depending on the extent of sequence editing (Fig 1B). The 938 genes upregulated by SKN-1t[NNNN] overlap much more with SKN-1t[NDNN] (749/938 = 80%) than with SKN-1t[DNDD] (235/938 = 25%). Although SKN-1t[DNDD] activates fewer genes in total (431), it has a large proportion (125/431 = 29%) of uniquely upregulated genes that are not similarly increased by either SKN-1t[NNNN] or SKN-1t[NDNN]. Thus, the extent to which the transcriptional output of SKN-1A is altered compared to that of non-edited SKN-1C is defined by the extent of sequence editing.

We were particularly struck by the identification of genes that are upregulated by greater than 2-fold in only one of the three SKN-1t transgenic strains (Fig 1B). This finding could imply the existence of unique transcriptional programs that are tied to specific patterns of sequence editing of SKN-1A. However, the 189 genes uniquely upregulated >2-fold

by SKN-1t[NNNN] are strongly skewed towards increased expression in the SKN-1t[DNDD] and SKN-1t[NDNN] transgenic animals (S2 Fig). Similarly, the 99 genes uniquely upregulated >2-fold by SKN-1t[NDNN] also show a bias towards increased expression in the other two transgenic strains (S2 Fig). More than 80% of the genes that are apparently uniquely upregulated by SKN-1t[NNNN] or SKN-1t[DNDD] are also upregulated in at least one other SKN-1t-expressing strain when using less stringent criteria (fold-change>1.5, FDR<0.05) (S2 Fig). These data argue against a unique program of transcriptional upregulation driven by SKN-1t[NNNN] or SKN-1t[NDNN], and rather suggest that the potency of some genes' activation is fine-tuned by low levels of sequence editing such that they fall just above or below the statistical cutoff used in our primary analysis. In contrast, the 125 genes that are uniquely upregulated >2-fold by SKN-1t[DNDD] correspond to a specific sequence editing-dependent transcriptional program. These genes show a bias towards increased expression in SKN-1t[NDNN]-expressing animals, but not in those expressing SKN-1t[NNNN] (S2 Fig). Thus, these data define a set of genes for which upregulation requires sequence editing and so are potentially regulated by SKN-1A but not by SKN-1C. Further, our data suggests that for these genes, strong activation by SKN-1A is contingent upon sequence editing at multiple Asn residues.

232 genes are upregulated by at least 2-fold by all three SKN-1t transgenes, suggesting they may be regulated by both SKN-1A and SKN-1C. On average, these genes are more strongly activated by SKN-1t[NNNN], most modestly upregulated by SKN-1t[DNDD], and the effect of SKN-1t[NDNN] is intermediate (Fig 1C). The 517 genes that are >2-fold upregulated by both SKN-1t[NNNN] and SKN-1t[NDNN] are on average more strongly activated by SKN-1t[NNNN] (S3 Fig), and skew towards upregulation in SKN-1t[DNDD]-expressing animals (S3 Fig). Collectively, these data suggest that sequence edited SKN-1A activates many of the same genes that are upregulated by non-edited SKN-1C, but does so less potently, especially if SKN-1A is sequence edited at multiple Asn residues. The 74 genes that are >2-fold upregulated by both SKN-1t[NDNN] and SKN-1t[DNDD] are more strongly upregulated by SKN-1t[DNDD] and less pronounced skew towards upregulation in SKN-1t[NNNN]-expressing animals (S3 Fig). This suggests that these genes are more potently activated by extensively sequence-edited SKN-1A and are not subject to regulation by non-edited SKN-1C. Taken together, these data suggest that the extent to which SKN-1A undergoes sequence editing sculpts its function in two ways. First, sequence-edited SKN-1A can upregulate genes that are also regulated by SKN-1C, but activation is dampened in a manner proportional to the number of Asn residues that undergo sequence editing. Second, sequence edited SKN-1A can activate another set of genes that are not regulated by SKN-1C. In this latter case, activation is potentiated in a manner proportional to the extent of sequence editing.

We therefore divided the SKN-1t-upregulated genes into three categories according to the effect of sequence editing on their activation (Fig 2A and S2 Table; see methods): (1) 'high-D' genes that are strongly (>2-fold) upregulated by sequence-edited SKN-1t only, likely to be uniquely under the control of sequence edited SKN-1A. (2) 'overlap' genes that are strongly upregulated by all three SKN-1t transgenics. These genes are potentially under the control of both SKN-1A and SKN-1C, although non-edited SKN-1C is a more potent activator. (3) 'low-D' genes that are strongly upregulated by non-edited and/or SKN-1t edited at a single Asn residue only. These genes are likely to be primarily controlled by SKN-1C but may potentially be regulated by partially sequence edited forms of SKN-1A.

## Sequence editing alters the functional profile of gene activation by SKN-1A

We used WormCat to define the functional impact of sequence editing [53]. In general, genes upregulated by SKN-1t transgenes are enriched for functional categories that correspond to known functions of *skn-1*, including pathogen responses, xenobiotic detoxification, metabolism, glutathione-S-transferases (GSTs), UDP-glucuronosyltransferases (UGTs), and proteasome subunits (Figs 2B and S4). Enrichment for proteasome subunit genes is only found in the SKN-1t[DNDD]-activated genes and is restricted to the 'high-D' category, suggesting that high levels of sequence editing are required for their activation (Figs 2B and S5). Interestingly, all proteasome subunit genes (except for *rpn-6.2*, which is a sperm-specific paralog of *rpn-6.1*) are consistently upregulated in both SKN-1t[NDNN] and SKN-1t[DNDD] transgenics

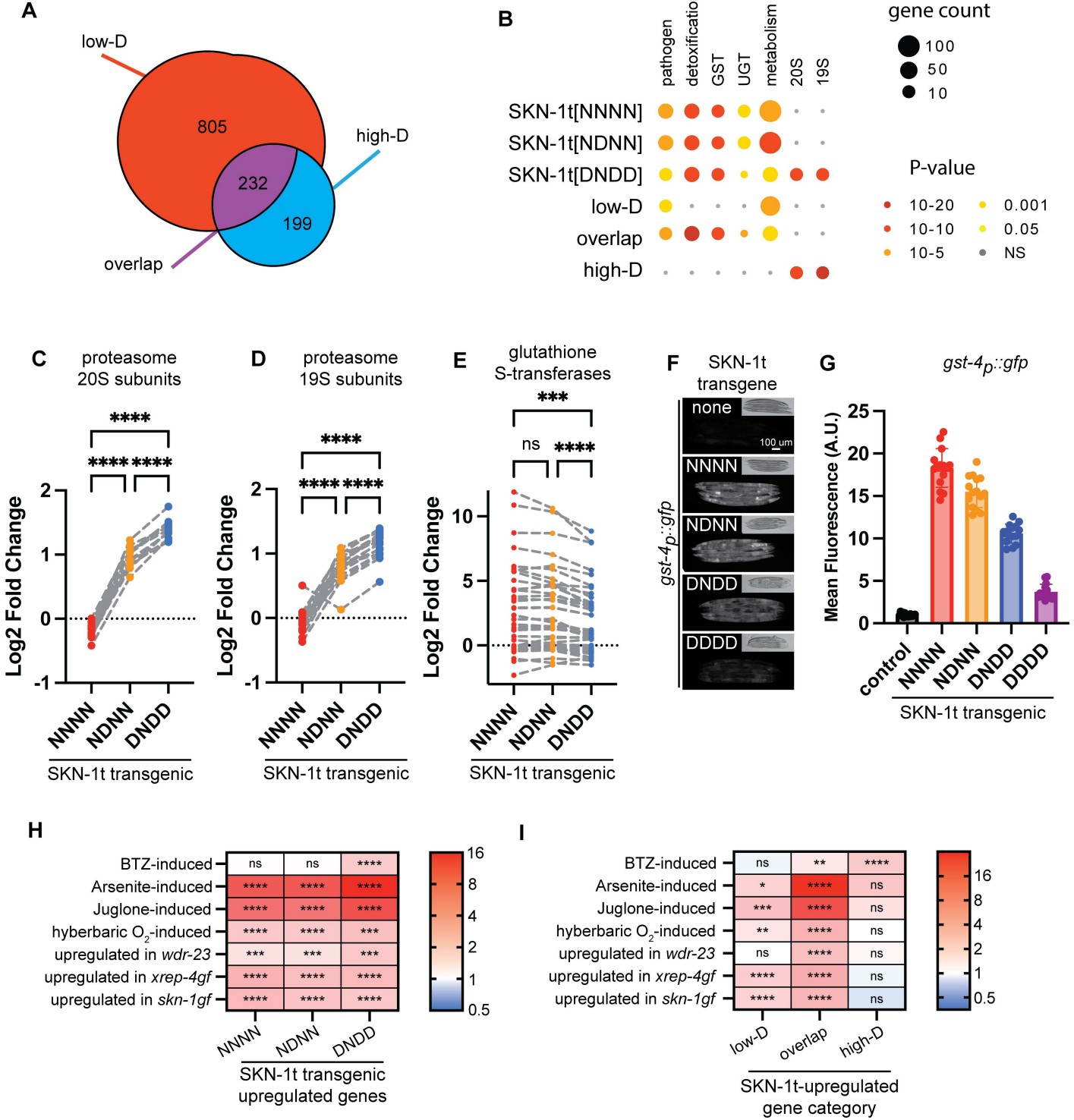

**Fig 2. Sequence editing alters the functional profile of gene activation by SKN-1A.** A) Classification of SKN-1t-upregulated genes into three classes according to the effect of sequence editing. The three classes are defined as follows: (1) 'high-D' genes are upregulated (>2-fold, FDR<0.01) in SKN-1t[DNDD], but not in SKN-1t[NNNN] transgenics; (2) 'overlap' genes are upregulated (>2-fold, FDR<0.01) in SKN-1t[NNNN], SKN-1t[NDNN] and SKN-1t[DNDD] transgenics; (3) 'low-D' genes are upregulated in SKN-1t[NNNN] and/or SKN-1t[NDNN], but not in SKN-1t[DNDD] transgenics. B) Worm-Cat analysis of SKN-1t upregulated genes, showing that these genes are enriched for functions associated with SKN-1. Enrichment of genes related

to proteasome functions (19S and 20S proteasome subcomplexes) is dependent on sequence-editing. Enrichment of genes related to other functional categories is not sequence-editing dependent. C) Upregulation of genes encoding proteasome 20S subunits by SKN-1t requires sequence editing. For each transgenic strain, the log2 Fold Change compared to the wild type (non-transgenic) control is plotted. Fold Change of each gene in different transgenic strains is connected by a dashed line. **** p<0.0001, repeated measures one-way ANOVA with Geisser-Greenhouse correction and Tukey's multiple comparisons test. D) Upregulation of genes encoding proteasome 19S subunits by SKN-1t requires sequence editing. For each transgenic strain, the log2 Fold Change compared to the wild type (non-transgenic) control is plotted. Fold Change of each gene in different transgenic strains is connected by a dashed line. **** p<0.0001, repeated measures one-way ANOVA with Geisser-Greenhouse correction and Tukey's multiple comparisons test. E) Upregulation of genes encoding glutathione-S-transferases by SKN-1t is fine-tuned by sequence editing. For each transgenic strain, the log2 Fold Change compared to the wild type (non-transgenic) control is plotted. Fold Change of each gene in different transgenic strains is connected by a dashed line. **** p<0.0001, *** p<0.001, ns p>0.05, repeated measures one-way ANOVA with Geisser-Greenhouse correction and Tukey's multiple comparisons test. F) Fluorescence micrographs showing that SKN-1t-driven transcriptional activation of the $gst-4_p::gfp$ reporter is modulated by mutations that mimic different degrees of sequence editing. Scale bar shows 100 μm. G) Quantification of fluorescence images shown in (F). $gst-4_p::gfp$ is most potently activated by SKN-1t[NNNN], mimicking activation of SKN-1C, and is most weakly activated by SKN-1t[N325D,N338D, N375D, N403D] (SKN-1t[DDDD]). n=15 animals measured per genotype. Error bars show mean±SD. The mean fluorescence of each strain is significantly different to that of every other strain, p<0.0001, ordinary one-way ANOVA with Tukey's multiple comparisons test. H and I) SKN-1t-upregulated genes are enriched for genes that are upregulated under diverse stresses and in mutants that display constitutive activation of skn-1-dependent stress responses. Enrichment of genes involved in specific stress responses is dependent on sequence editing. Color indicates the enrichment score, calculated as the ratio of the number of overlapping genes over the number of overlapping genes expected by chance. An enrichment score >1 indicates more overlap than expected from two independent gene sets. **** p<$10^{-10}$, *** p<$10^{-5}$, ** p<0.001, * p<0.01, ns p>0.01, hypergeometric test. Numerical data for panel G are available in S1 Data.

(Fig 2C and 2D). The upregulation of proteasome subunits by SKN-1t[NDNN] falls just below the 2-fold change cut-off used in our analysis, explaining the lack of enrichment. These data indicate an essential role for extensive sequence editing in SKN-1A/Nrf1-mediated activation of proteasome biogenesis. The more potent activation of proteasome subunits by more highly sequence-edited SKN-1t is consistent with our previous findings using a proteasome subunit transcriptional reporter [21].

Genes associated with detoxification and oxidative stress defense functions are concentrated in the 'overlap' category, suggesting sequence-edited SKN-1A and non-edited SKN-1C are both competent to activate oxidative stress response/detoxification genes (Figs 2B and S5). GSTs are important effectors of these responses, and the same subset of GST genes is upregulated in all three transgenic strains (Fig 2E). Whilst each GST gene is induced to a similar extent in each SKN-1t-expressing strain, the average extent of induction in the SKN-1t[DNDD]-expressing animals is reduced compared to the other two strains (Fig 2E). This finding matches the generally attenuated activation of genes in the 'overlap' category by SKN-1t[DNDD] (Fig 1C). Thus, these data suggest that sequence editing attenuates the extent of activation of oxidative stress responsive genes by SKN-1A. We confirmed this effect with a GFP reporter for the transcription of gst-4 ($gst-4_p::gfp)$, which is induced in a skn-1-dependent manner during oxidative stress [39,54] (Fig 2F and 2G). These results suggest that sequence editing fine tunes SKN-1A function by differentially adjusting the upregulation of genes across different functional classes.

## Sequence editing adjusts gene activation by SKN-1 isoforms to mediate distinct stress responses

We next examined the relationship between sequence editing and stress-responsive gene expression programs linked to skn-1. First, we compared different classes of SKN-1t-activated genes to genes that are induced in animals exposed to the proteasome inhibitor bortezomib (BTZ), which triggers SKN-1A-dependent activation of proteasome subunit gene expression [20,55]. BTZ-induced genes are highly enriched amongst the genes upregulated by SKN-1t[DNDD]-expressing animals, but not the other transgenic strains (Fig 2H). This suggests that highly sequence-edited forms of SKN-1A drive transcriptional adaptation to proteasome inhibition. Additionally, BTZ-induced genes are not enriched in the 'low-D' category (Fig 2I), arguing against strong BTZ-induced activation of SKN-1A that has undergone low levels of sequence editing, or activation of SKN-1C. Interestingly, BTZ-induced genes are enriched in the 'overlap' category (Fig 2I), suggesting that during proteasomal inhibition, genes typically activated by SKN-1C (see below) may be under the transcriptional control of

SKN-1A. Consistently, *gst-4*, a gene in the overlap category, is induced in a *skn-1a* and *png-1*-dependent manner following BTZ exposure [21]. Strikingly, the genes from the 'high-D' category that are also induced by WT animals in response to BTZ include almost all proteasome subunits and many other proteins implicated in UPS function (S3 Table). Thus, these data indicate the transcriptomic basis for the essential role of sequence editing of SKN-1A in proteasome inhibitor resistance.

Genes induced under oxidative stress conditions (10 mM sodium arsenite, 38 uM juglone, or hyperbaric oxygen) are highly enriched in the genes upregulated in all three SKN-1t transgenic strains [56–58] (Fig 2H). But critically, when SKN-1t-upregulated genes are divided according to sensitivity to sequence editing, there is only enrichment in the 'overlap' and 'low-D' categories, not the 'high-D category' (Fig 2I). A similar pattern of enrichment is present among genes activated by animals harboring mutations that cause hyperactivation of *skn-1*-dependent oxidative stress responses [59,60] (Fig 2H and 2I). The lack of enrichment in the 'high-D' category argues that non-edited SKN-1C is the primary driver of the oxidative stress response. Genes from the 'overlap' category that are induced by WT animals in response to multiple oxidants include many with putative oxidative stress and detoxification related functions (S4 Table). Strikingly, most of these genes are also induced in animals exposed to BTZ (S4 Table). Thus, although our data imply that they are driven by different forms of SKN-1, the transcriptional responses to BTZ and to oxidative stress do overlap.

Collectively, these data suggest that different forms of SKN-1 can drive upregulation of overlapping transcriptional programs that are tailored for different stress conditions. In a straightforward model, sequence-edited SKN-1A mediates the response to proteasome inhibition, and (non-edited) SKN-1C drives the response to oxidative stress. However, the transcriptional output of sequence edited SKN-1A overlaps with that of SKN-1C to include many genes implicated in oxidative stress responses. As a result, these data also raise the possibility that sequence edited SKN-1A and SKN-1C both contribute to oxidative stress responses.

### Isoform-specific functions of SKN-1A and SKN-1C

To address the possible overlapping functions of SKN-1A and SKN-1C, we sought to compare the phenotypic effects of genetic inactivation of each isoform in isolation, or both in combination. To analyze the effect of eliminating SKN-1C with minimal alteration to SKN-1A levels or function, we replaced the initiator ATG of the SKN-1C isoform open reading frame (ORF) with GCT, which encodes alanine and cannot serve to initiate translation (Fig 3A). There are no nearby in-frame start codons that might be used as alternative translation initiation sites in animals lacking the canonical SKN-1C start codon, so this mutation is likely to eliminate SKN-1C expression. The initiator methionine of SKN-1C is also the 91st amino acid of the SKN-1A ORF. Thus, animals harboring this engineered mutation are expected to lack SKN-1C and produce a mutant form of SKN-1A harboring an M91A amino acid substitution. We generated this mutation (the *nic952* allele) by CRISPR/Cas9 gene editing and confirmed the correct edit by Sanger sequencing. For simplicity, we hereafter refer to animals harboring the *skn-1(nic952*[M1A/M91A]) allele as *skn-1c* mutants. We will refer to animals harboring the *skn-1(mg570)* allele, which introduces a stop codon to a *skn-1a*-specific exon [20], as *skn-1a* mutants; and animals harboring the *skn-1(zu67)* allele, which introduces a premature stop codon into an exon shared by *skn-1a* and *skn-1c* coding sequences [61], as *skn-1ac* mutants (Fig 3A).

The *skn-1ac* mutation causes maternal effect embryonic lethality [62,63]. Notably, this maternal effect lethality is shared with *skn-1c* mutants, whereas *skn-1a* mutants do not show any sign of embryonic lethality (Fig 3B). We conclude that the elimination of the SKN-1C initiator methionine abrogates an essential function of a maternally derived *skn-1* gene product in embryonic development. WDR-23 is thought to prevent constitutive activation of *gst-4* by inhibiting SKN-1C [39]. We tested the effect of abrogation of the *skn-1c* initiator methionine on constitutive activation of a *gst-4$_p$::mCherry* reporter in a *wdr-23* null mutant. Strikingly, hyperactivation of *gst-4$_p$::mCherry* is lost in *skn-1c* and *skn-1ac* mutants but is unaltered in animals lacking SKN-1A (Fig 3C and 3D). We conclude that elimination of the SKN-1C initiator methionine abrogates upregulation of *gst-4* caused by loss of WDR-23.

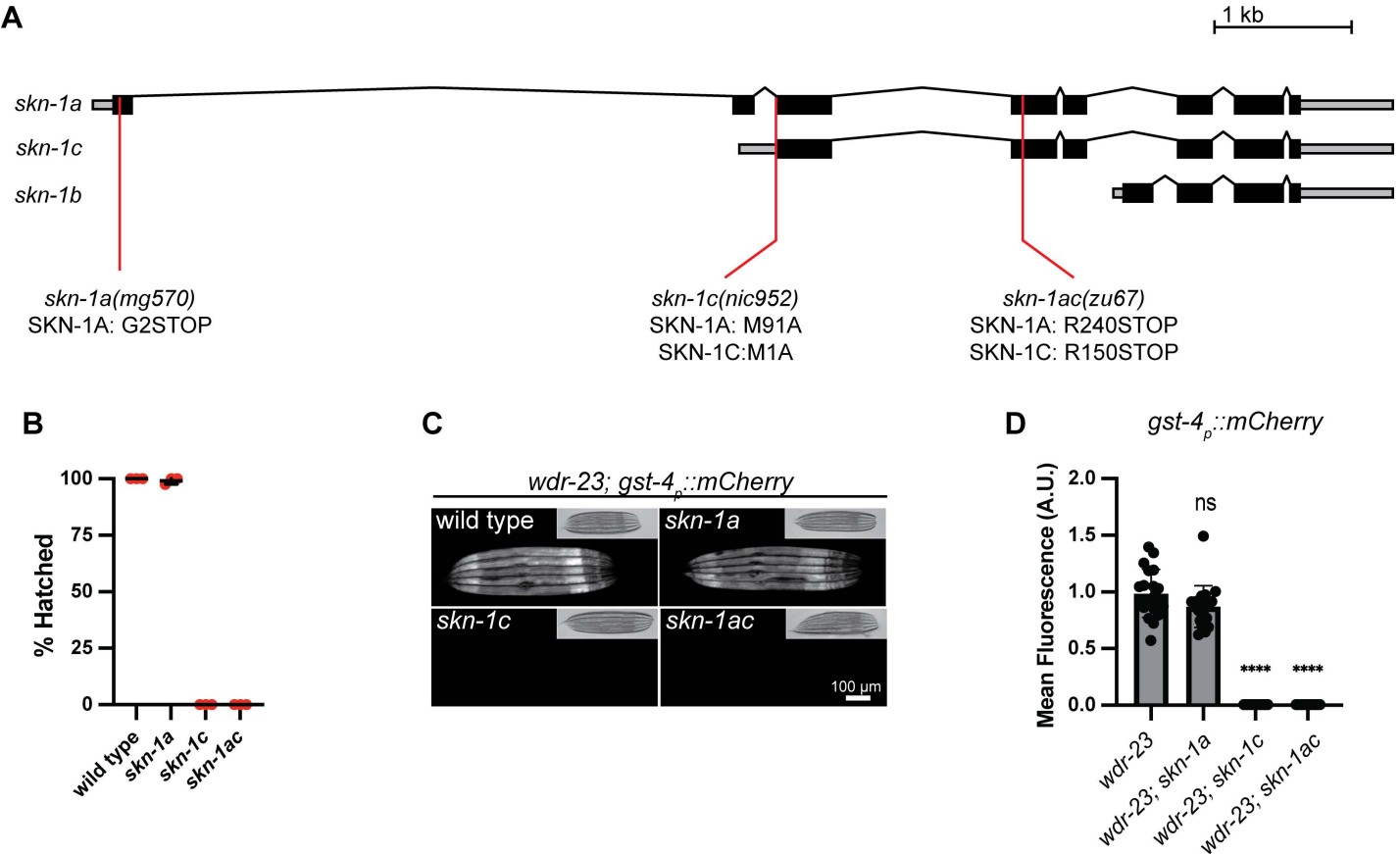

**Fig 3. Removal of the initiator methionine of the SKN-1C isoform impairs some *skn-1* functions.** A) Schematic showing the *skn-1* locus and the locations of mutations used in this study. *mg570* is a premature termination codon specific to the *skn-1a* coding sequence [G2STOP]. The *nic952* allele replaces the initiator methionine of the *skn-1c* open reading frame with an alanine codon [M1A], also altering the 91st codon of the *skn-1a* coding sequence [M91A]. *zu67* is a premature termination codon that affects both the *skn-1a* and *skn-1c* coding sequences [R240STOP and R150STOP, respectively]. B) Abrogation of the *skn-1c* initiator methionine causes a maternal-effect lethal phenotype. Error bars show mean±SD. Results of n=3 replicate experiments are shown. Hatching of at least 50 embryos was scored for each replicate. C) Fluorescence images showing the effect of *skn-1* mutations on hyperactivation of *gst-4_p::mCherry* in animals lacking WDR-23. Hyperactivation of *gst-4_p::mcherry* does not require SKN-1A, but is abrogated in animals lacking SKN-1C. Scale bar shows 100 µm. D) Quantification of the effect of *skn-1* mutations on hyperactivation of *gst-4_p::mCherry* in animals lacking WDR-23. n=20 animals per genotype. Error bars show mean±SD. **** p<0.0001, ns p>0.05, ordinary one-way ANOVA with Tukey's multiple comparisons test. Numerical data for panels B and D are available in S1 Data.

These data suggest that the SKN-1C isoform non-redundantly performs some *skn-1* functions. However, the *skn-1c* mutation also affects SKN-1A (via the M91A amino acid substitution). As such, these data do not exclude the possibility that SKN-1A[M91A] is non-functional, and the two SKN-1 isoforms act redundantly in embryonic development or downstream of WDR-23. Animals lacking SKN-1A are defective in regulation of proteasome subunit gene expression and are highly sensitive to killing by the proteasome inhibitor BTZ [20,21]. To test whether the M91A amino acid substitution interferes with SKN-1A function, we examined the sensitivity of *skn-1c* mutants to BTZ. We confirmed that both *skn-1a* and *skn-1ac* mutants are highly sensitive to BTZ-mediated killing and growth inhibition as expected. However, *skn-1c* mutants are not sensitive, instead they exhibit survival and growth comparable to wild type animals (Fig 4A and 4B). Consistently, both the basal expression level and the BTZ-responsive upregulation of the proteasome subunit gene expression *rpt-3_p::gfp* is unchanged in *skn-1c* mutant animals compared to the wild type (Fig 4C and 4D). Further, *skn-1c* mutants,

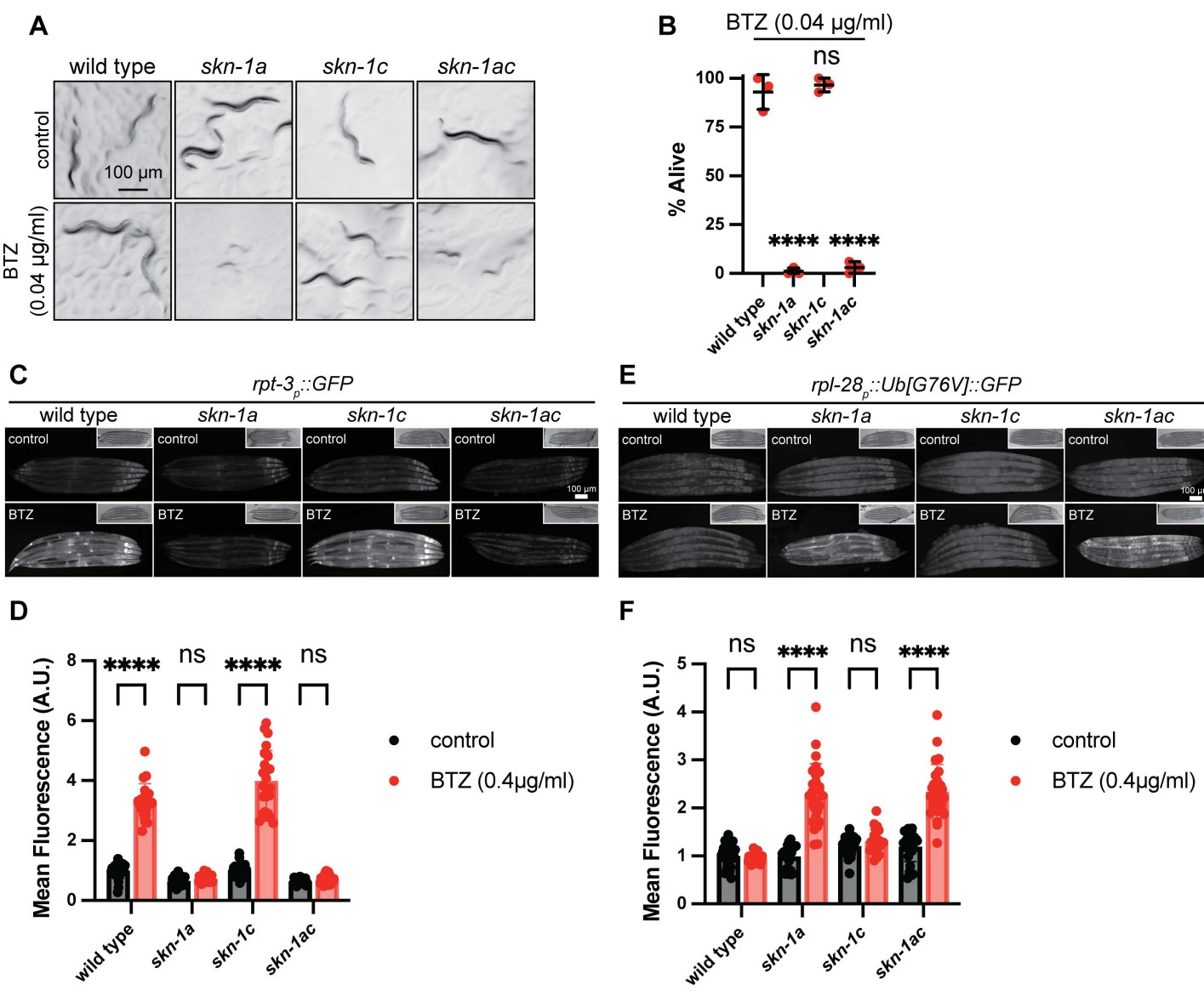

**Fig 4. Removal of the initiator methionine of the SKN-1C isoform does not impact SKN-1A functions.** A) Images showing that abrogation of the *skn-1c* initiator methionine does not cause hypersensitivity to growth inhibition by bortezomib. 5-10 L4 animals were shifted to control (DMSO) or 0.04 µg/ml (104 nM) bortezomib-supplemented plates and the growth of their progeny imaged after 3 days. *skn-1a* and *skn-1ac* mutant animals show early larval arrest/lethality, whereas *skn-1c* animals develop similarly to the wild type. Scale bar shows 100 µm. B) Abrogation of the *skn-1c* initiator methionine does not reduce survival of adult animals exposed to bortezomib. Late L4 stage animals of each genotype were shifted to bortezomib supplemented plates (0.04 µg/ml, 104 nM) and checked for survival after 4 days. Results of n = 3 replicates are shown. Survival of 30 animals was tested for each replicate. Error bars show mean ± SD. **** p < 0.0001, ns p > 0.05 (p-values comparing mean survival of each mutant to the wild-type control), ordinary one-way ANOVA with Dunnett's multiple comparisons test. C) Fluorescence micrographs showing induction of the *rpt-3$_p$::gfp* proteasome subunit reporter following bortezomib exposure. Animals were transferred to plates supplemented with 0.4 µg/ml BTZ for 20 hours prior to imaging. Abrogation of the *skn-1c* initiator methionine does not reduce stress-responsive induction of the reporter. Scale bar shows 100 µm. D) Quantification of *rpt-3$_p$::gfp* induction in *skn-1* mutants shown in (C). n > 20 animals were measured per genotype/condition. Error bars show mean ± SD. **** p < 0.0001, ns p > 0.05, ordinary two-way ANOVA with Sidak's multiple comparisons test. E) Fluorescence micrographs showing accumulation of the proteasome substrate Ub[G76V]::GFP following exposure to the proteasome inhibitor bortezomib. Animals were raised at 25°C to the L4 stage, and then transferred to plates supplemented with 0.4 ug/ml BTZ for 20 hours prior to imaging. Abrogation of the *skn-1c* initiator methionine does not compromise animals' ability to maintain proteasome function following bortezomib challenge. Scale bar shows 100 µm. F) Quantification of Ub[G76V]::GFP in *skn-1* mutants shown in (E). n > 20 animals were measured per genotype/condition. Error bars show mean ± SD. **** p < 0.0001, ns p > 0.05, ordinary two-way ANOVA with Sidak's multiple comparisons test. Numerical data for panels B, D, and F are available in S1 Data.

unlike *skn-1a* or *skn-1ac* mutant animals, do not show any defect in proteasome function as measured by degradation of unstable Ub[G76V]::GFP [64–66] (Figs 4E, 4F and S6). In sum, proteasome regulation and function are normal in the *skn-1c* mutants, indicating that the M91A amino acid substitution does not disrupt SKN-1A function. We conclude that the phenotypic consequences of the *skn-1c* mutation result from inactivation of SKN-1C.

Numerous studies have indicated critical roles for *skn-1* in normal lifespan and longevity through regulation of various cellular and organismal processes [67–70]. The lifespan of animals lacking SKN-1A is reduced to the same extent as those lacking both SKN-1A and SKN-1C [64]. Interestingly, the lifespan of *skn-1c* mutants is unaltered compared to the wild type (Fig 5A and S5 Table). Additionally, *skn-1ac* and *skn-1a* mutants both display an accelerated age-related vulval integrity defect (the Avid phenotype) compared to the wild type, indicative of an age-dependent defect in tissue homeostasis [64,71]. However, the *skn-1c* mutation does not have this effect (Fig 5B). These data indicate that SKN-1C is not required for normal lifespan or to maintain tissue homeostasis during aging.

Together, these data indicate that SKN-1A and SKN-1C perform at least some separate and non-redundant functions. SKN-1A is required for some processes that are unaffected in animals lacking SKN-1C: (1) proteasome regulation, and (2) normal lifespan and maintenance of tissue homeostasis during aging. In contrast, SKN-1C is required for some processes that are unaffected in animals lacking SKN-1A: (1) embryogenesis, and (2) hyperactivation of *gst-4* caused by loss of WDR-23. We generated transcriptional reporters and confirmed that SKN-1A and SKN-1C isoforms are both constitutively expressed in ~all somatic tissues (S7 Fig). Thus, these functional distinctions between the two isoforms are unlikely to be explained by differences in their expression patterns but instead reflect distinct mechanisms of activation and/or target specificity.

## SKN-1A and SKN-1C are both required for optimal oxidative stress resistance

Mutations or RNAi conditions that simultaneously ablate SKN-1A and SKN-1C render animals highly sensitive to oxidative stress [32,33,72,73]. Transgenic expression of SKN-1C can rescue this defect, but overexpression of SKN-1C from

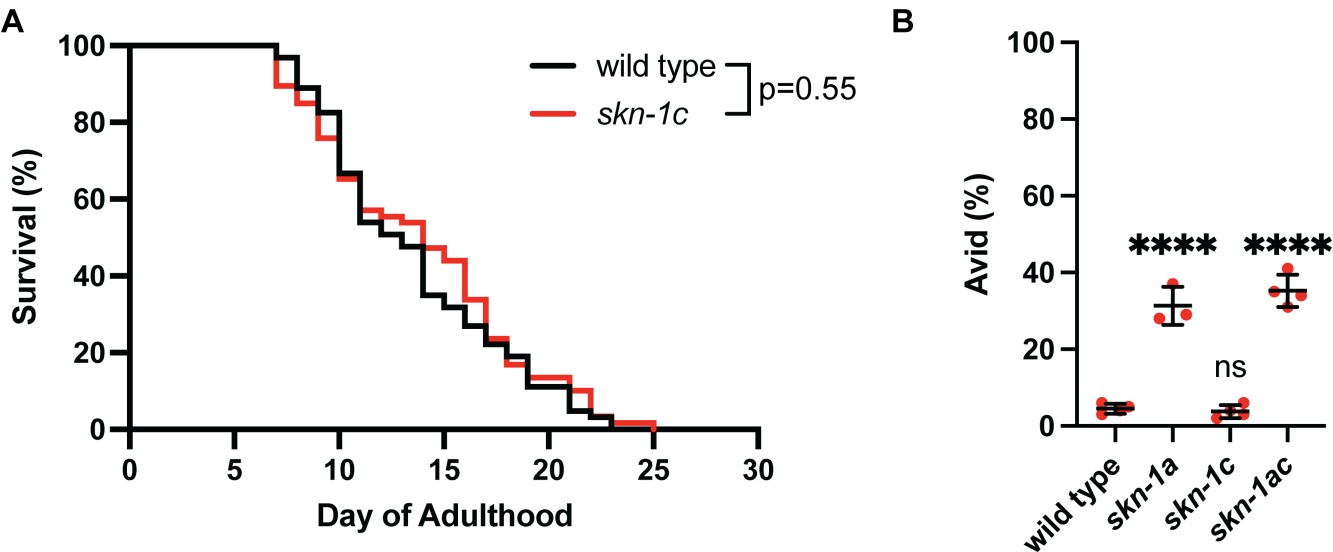

**Fig 5. SKN-1C is not required for normal lifespan or tissue integrity during aging.** A) Survival curve showing that the lifespan *of skn-1c* mutant animals is not reduced compared to the wild type. p = 0.55, Log-rank (Mantel-Cox) test. Survival of n > 60 animals was measured for each genotype. B) Analysis of vulval degeneration in day 7 adults. *skn-1a* and *skn-1ac*, but not the *skn-1c* mutation, cause increased age-associated vulval degeneration. n = 3 or n = 4 replicate experiments are shown. Age-dependent vulval degeneration of at least 50 animals was assayed for each replicate. Error bars show mean ± SD. **** p < 0.0001, ns p > 0.05, ordinary one-way ANOVA with Dunnett's multiple comparisons test. Numerical data for panel B are available in S1 Data.

a transgene may mask any contribution of SKN-1A [33,73]. We therefore sought to clarify the individual contributions of endogenous SKN-1A and SKN-1C by comparing the *skn-1a*, *skn-1c* and *skn-1ac* mutants' ability to withstand oxidative stress.

Arsenite (AS) is reactive to thiol groups of proteins and glutathione and stimulates reactive oxygen species (ROS) production, all of which contribute to oxidative stress [74]. Arsenite exposure leads to *skn-1*-dependent activation of anti-oxidant and detoxification gene expression and *skn-1* mutants that lack both the SKN-1A and SKN-1C isoforms are highly sensitive to killing by arsenite [33]. We confirmed that *skn-1ac* mutants are profoundly sensitive to arsenite; the survival of *skn-1ac* animals is significantly reduced compared to the wild type at all three AS concentrations tested (1–3 mM) (Fig 6A–6C). Almost all *skn-1ac* animals die within 24 hours of exposure to 2mM or 3mM arsenite, even though almost all wild type animals survive. The reduction in survival of animals lacking *skn-1a* compared to the wild type was not statistically significant at any concentration, whereas *skn-1c* mutants showed a significant reduction in survival compared to wild type when exposed to 2 mM or 3mM arsenite (Fig 6A–6C). These results suggest that endogenous SKN-1C, but not SKN-1A, is sufficient for normal arsenite resistance under these conditions. Nonetheless, *skn-1ac* mutants have lower survival rates than animals lacking only SKN-1C at both 2mM and 3mM arsenite, indicating that endogenous SKN-1A contributes to arsenite resistance in the absence of SKN-1C (Fig 6B and 6C).

Paraquat (PQ) is highly toxic, causes ROS production and oxidative stress [75]. Like AS, PQ triggers a *skn-1*-dependent transcriptional response, and *skn-1* mutants are more sensitive than the wild type to killing by PQ [32]. We measured animals' development in the presence of 4 mM PQ, a concentration that we found causes a severe developmental delay in wild type animals (Fig 6D). Strikingly, the developmental rate of *skn-1a* or *skn-1c* mutants exposed to PQ was similar to that of wild-type animals, whereas the *skn-1ac* mutants were significantly delayed (Fig 6D). This indicates that SKN-1A and SKN-1C redundantly confer oxidative stress resistance in the context of PQ exposure during development. Although the relative contribution of each isoform can clearly differ depending on the type of oxidative stress applied, collectively, these data reveal that both SKN-1A and SKN-1C are required for optimal oxidative stress resistance. We attempted but were not able to establish robust assays for the effect of AS on developmental progression, or the effect of PQ on adult survival.

## SKN-1A and SKN-1C have distinct functions in regulation of gene expression during oxidative stress

*gst-4$_p$::gfp*, is activated in a *skn-1*-dependent manner during oxidative stress [39,76]. As expected, *gst-4$_p$::gfp* induction following either AS or PQ exposure is completely abrogated in the *skn-1ac* mutants (Fig 6E–6L). In the isoform-specific mutants, *gst-4$_p$::gfp* induction in response to AS is primarily dependent on SKN-1C, although SKN-1A contributes to full activation of the response (Fig 6E–6H). The reporter is weakly induced after 20 hours of AS exposure in *skn-1c* mutants in a way that requires SKN-1A, as this induction is absent in the *skn-1ac* mutants (Fig 6F and 6H). *gst-4$_p$::gfp* induction in animals exposed to PQ is completely lost in animals lacking SKN-1C (Fig 6I–6L). There is a significant reduction in induction in *skn-1a* mutants following 4 hours PQ exposure, suggesting that SKN-1A may play a role in ensuring rapid *gst-4$_p$::gfp* induction (Fig 6I and 6K). These results suggest that SKN-1A and SKN-1C can both contribute to the regulation of *gst-4::gfp* expression during oxidative stress. *gst-4$_p$::gfp* expression is also induced when proteasome function is impaired [76], but in that context, *skn-1a* is essential [21]. Further, XREP-4, which promotes *gst-4$_p$::gfp* activation under oxidative stress, is not required for *gst-4$_p$::gfp* activation in the context of proteasome impairment [77]. Thus, although SKN-1A and SKN-1C can both upregulate the *gst-4$_p$::gfp* reporter, their relative contributions are context-specific.

## The peptide:N-glycanase PNG-1 is required for SKN-1A-dependent oxidative stress defenses

The peptide:N-glycanase enzyme PNG-1/NGLY1 is required to deglycosylate and sequence edit SKN-1A after its release from the ER [20,21]. To examine the role of sequence editing of SKN-1A in oxidative stress responses, we used animals harboring a null allele affecting *png-1* (the *ok1654* allele contains a ~1.1 kb deletion that removes part of the

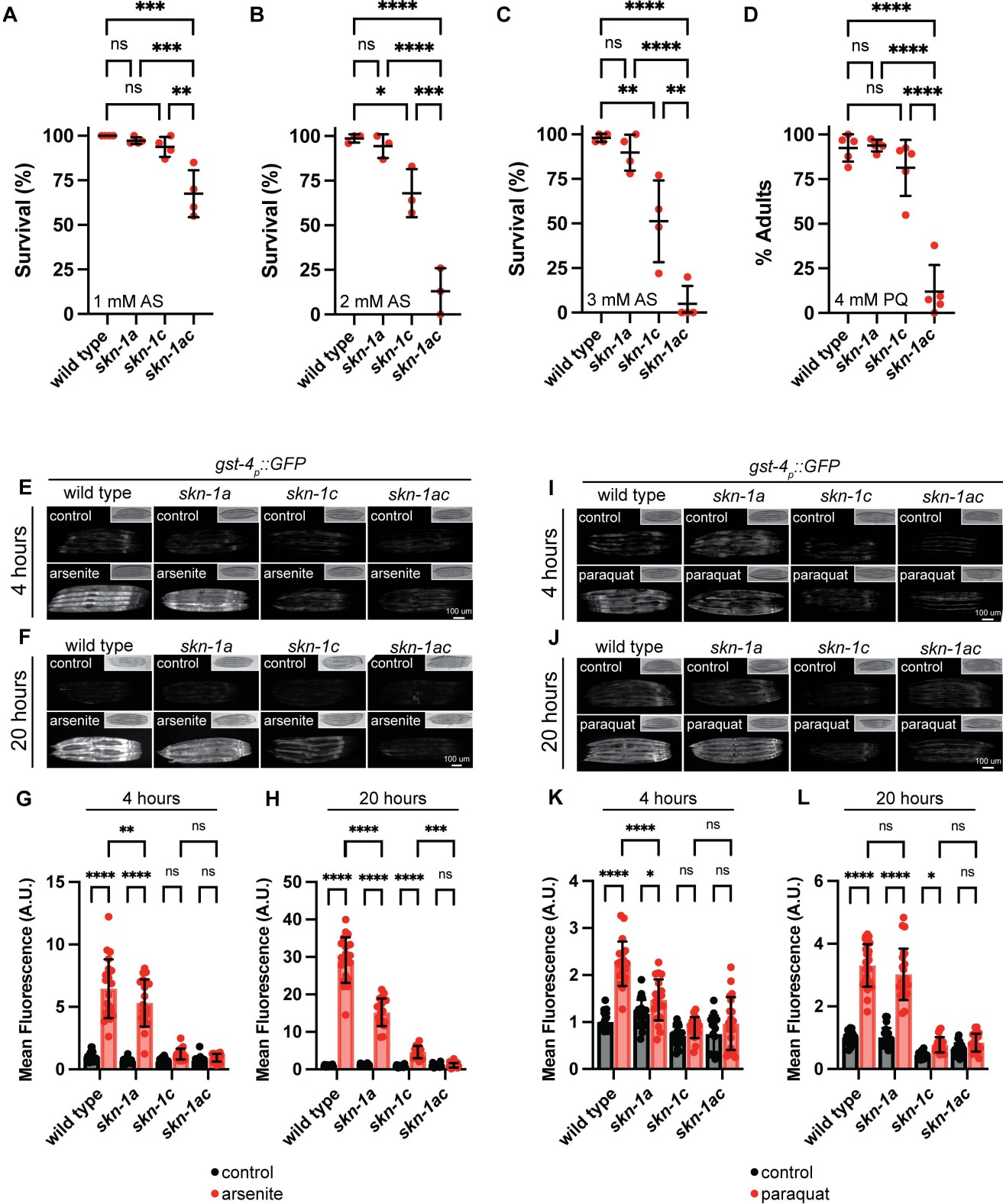

**Fig 6. SKN-1A and SKN-1C are both required for optimal oxidative stress resistance.** A-C) Survival of animals exposed to 1-3 mM arsenite. Late L4 stage animals of each genotype were shifted to arsenite supplemented plates and checked for survival after 24 hours. Results of n = 3 replicates are shown, survival of at least 20 animals was tested for each replicate. Error bars show mean ± SD. **** p < 0.0001, *** p < 0.001, ** p < 0.01, * p < 0.05, ns

p > 0.05, ordinary one-way ANOVA with Tukey's multiple comparisons test. D) Development of animals in the presence of 4 mM paraquat showing that *skn-1a* and *skn-1c* are redundantly required for development in the presence of paraquat. Development was scored 7 days after synchronized egg lay. Results of n = 5 replicates are shown, at least 35 animals' development was assayed for each replicate. Error bars show mean ± SD. **** p < 0.0001, ns p > 0.05, ordinary one-way ANOVA with Šídák's multiple comparisons test. E, F) Fluorescence micrographs showing that *skn-1a* and *skn-1c* both contribute to full activation of *gst-4$_p$::gfp* following arsenite exposure (2 mM, 4-20 hours). Scale bar shows 100 μm. G, H) Quantification of *gst-4$_p$::gfp* induction in *skn-1* mutants exposed to arsenite as shown in (E, F). n > 19 animals per genotype/condition. Error bars show mean ± SD. **** p < 0.0001, *** p < 0.001, ** p < 0.01, ns p > 0.05, ordinary two-way ANOVA with Sidak's multiple comparisons test. I, J) Fluorescence micrographs showing induction of *gst-4$_p$::gfp* following paraquat exposure (3 mM, 4-20 hours). Activation of *gst-4$_p$::gfp* under this condition is lost in *skn-1c* but not *skn-1a* mutants. Scale bar shows 100 μm. K, L) Quantification of *gst-4$_p$::gfp* induction in *skn-1* mutants exposed to paraquat. as shown in (I, J). n > 19 animals per genotype/condition. Error bars show mean ± SD. **** p < 0.0001, * p < 0.05, ns p > 0.05, ordinary two-way ANOVA with Sidak's multiple comparisons test. Numerical data for panels A-D, G, H, K and L are available in S1 Data.

transglutaminase domain, we will hereafter refer to animals harboring this deletion as *png-1* mutants) [78]. When exposed to AS, *png-1* mutants show a phenotype consistent with inactivation of SKN-1A-dependent oxidative stress defenses: *png-1* single mutants are weakly sensitive to AS, whereas *png-1; skn-1c* double mutants are extremely sensitive, and phenocopy the *skn-1ac* mutant (Fig 7A). Similarly, *png-1* mutants exposed to PQ display a phenotype consistent with SKN-1A inactivation: *png-1* single mutants develop at the same rate as wild type animals in the presence of PQ, while *png-1; skn-1c* double mutants show severe developmental delay, phenocopying *skn-1ac* mutant animals (Fig 7B). We conclude that removal of N-glycans from SKN-1A by PNG-1 is necessary for optimal oxidative stress defenses. Possibly, SKN-1A must become sequence edited in order to promote oxidative stress defenses, but it is also possible that retained N-glycans interfere with transcriptional control by SKN-1A. The fact that the SKN-1C isoform promotes oxidative stress resistance without undergoing N-glycosylation or sequence editing supports the second possibility. In any case, because removal of N-glycans and sequence editing occur concomitantly, these data demonstrate that a sequence edited form of SKN-1A acts to promote optimal oxidative stress defenses together with non-edited SKN-1C.

### Distinct pathways govern oxidative stress resistance through SKN-1A and SKN-1C

Oxidative stress responses are constitutively activated in animals lacking WDR-23, which acts as a substrate receptor in a cullin-RING ligase complex to negatively regulate a *skn-1*-mediated oxidative stress response [39,40]. The effect of *wdr-23* inactivation on *gst-4* expression requires SKN-1C but not SKN-1A (Fig 3C and 3D). Under oxidative stress conditions, F-box protein XREP-4 inhibits WDR-23 to trigger stress response activation [77,79]. Thus, XREP-4 and WDR-23 likely constitute a signaling cascade that governs oxidative stress resistance through SKN-1C. To test this model, we generated a null allele of *xrep-4* (E22STOP; these animals are hereafter referred to as *xrep-4* mutants). We find that animals harboring the *xrep-4* mutation show impaired, yet not abolished, stress responsive activation of *gst-4$_p$::gfp* (S8 Fig). *xrep-4* is not required for upregulation of the proteasome subunit reporter *rpt-3$_p$::gfp* following BTZ exposure, indicating that XREP-4 is not essential for SKN-1A function (S8 Fig). In AS sensitivity assays, we found that loss of XREP-4 increases AS sensitivity of animals lacking SKN-1A or PNG-1 consistent with the participation of XREP-4 and SKN-1A/PNG-1 in separate pathways that independently promote oxidative stress resistance (Fig 7C). In contrast, XREP-4 inactivation does not further increase AS sensitivity of *skn-1c* mutants (Fig 7D). Thus, XREP-4/WDR-23 signaling governs activation of oxidative stress defenses via SKN-1C, not through regulation of sequence-edited SKN-1A. We conclude that distinct signaling pathways regulate SKN-1A and SKN-1C to precisely orchestrate activation of their shared transcriptional outputs and promote oxidative stress resistance.

## Discussion

### Sequence editing sculpts SKN-1A/Nrf1 function

SKN-1A and SKN-1C function analogously to Nrf1 and Nrf2 in the control of cellular homeostasis and stress responses [11,34]. Sequence editing, a post-translational modification unique to ER-associated SKN-1A/Nrf1, is essential for

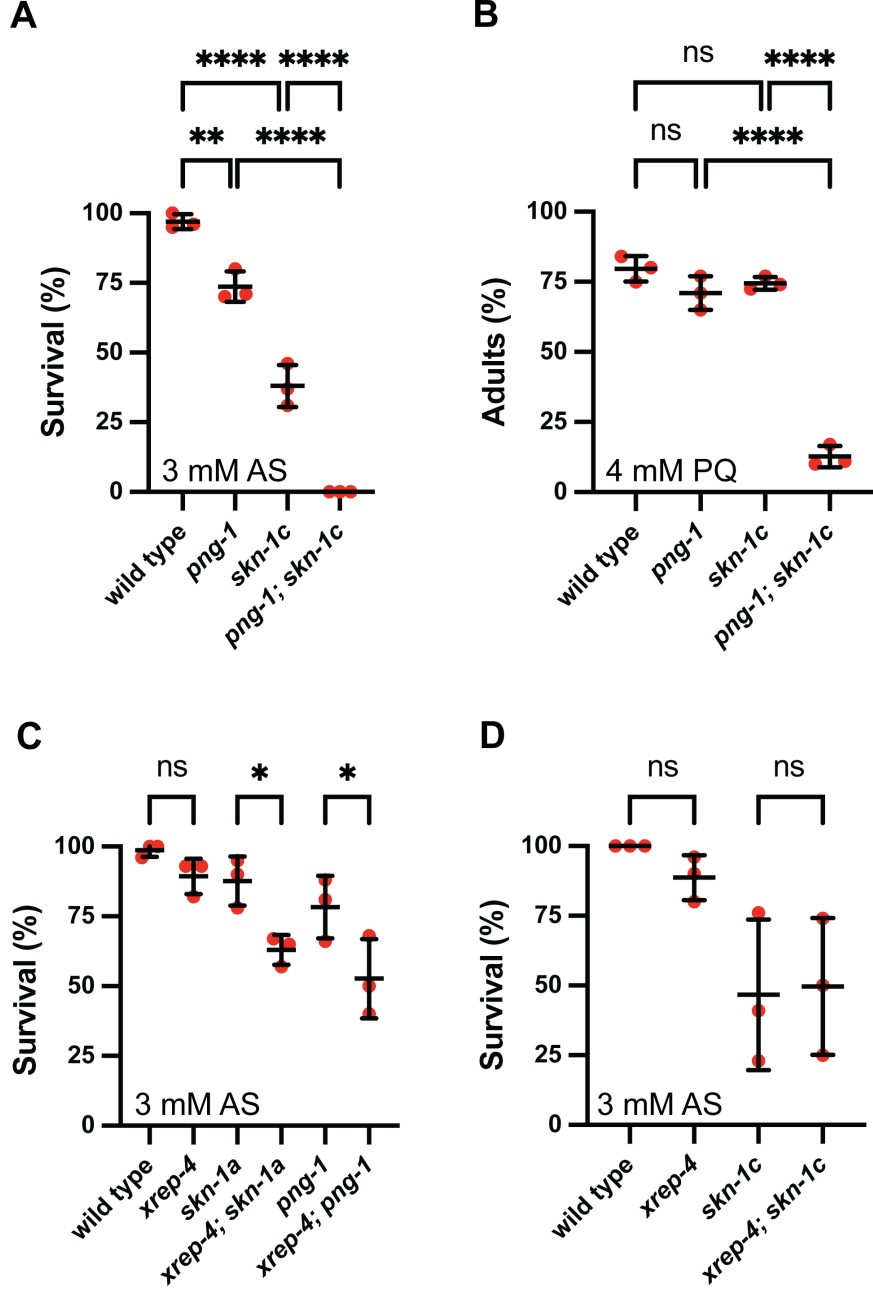

**Fig 7. Distinct mechanisms control oxidative stress resistance through SKN-1A and SKN-1C.** A) Survival of animals exposed to 3 mM arsenite showing that inactivation of *png-1* reduces the survival of wild-type and *skn-1c* mutant animals. Late L4 stage animals of each genotype were shifted to arsenite supplemented plates and checked for survival after 24 hours. Results of n = 3 replicates are shown, survival of at least 15 animals was tested for each replicate. Error bars show mean ± SD. **** p < 0.0001, ** p < 0.01, ordinary one-way ANOVA with Tukey's multiple comparisons test. B) Development of animals in the presence of 4 mM paraquat showing that *png-1* is required redundantly with *skn-1c* for development in the presence of paraquat. Development was scored 7 days after synchronized egg lay. Results of n = 5 replicates are shown, development of at least 30 animals was tested for each replicate. Error bars show mean ± SD. **** p < 0.0001, ns p > 0.05, ordinary one-way ANOVA with Tukey's multiple comparisons test. C) Inactivation of *xrep-4* reduces survival of *skn-1a* and *png-1* mutant animals in the presence of 3 mM arsenite. Late L4 stage animals of each genotype were shifted to arsenite supplemented plates and checked for survival after 24 hours. Results of n = 3 replicates are shown, survival of at least 15 animals was tested for each replicate. Error bars show mean ± SD. * p < 0.05, ns p > 0.05, ordinary one-way ANOVA with Tukey's multiple comparisons test. D) Inactivation of *xrep-4* does not alter survival of *skn-1c* mutant animals in the presence of 3 mM arsenite. Late L4 stage animals of each genotype were shifted to arsenite supplemented plates and checked for survival after 24 hours. n = 3 replicates are shown, survival of at least 15 animals was tested for each replicate. Error bars show mean ± SD. ns p > 0.05, ordinary one-way ANOVA with Tukey's multiple comparisons test. Numerical data for panels A-D are available in S1 Data.

SKN-1A/Nrf1-dependent regulation of proteasome biogenesis. However, its impact on other aspects of SKN-1A/Nrf1 function was unknown [21,22]. To investigate this, we performed transcriptome analysis of animals engineered to express hyperactive truncated SKN-1, with or without mutations that mimic the effect of deglycosylation-dependent sequence editing. The transgenic animals exhibited increased expression of genes involved in stress responses, proteostasis, metabolism, and aging (Figs 1, 2 and S1–S4 and S1–S4 Tables). The upregulated genes significantly overlap with those induced by endogenous SKN-1A/C under stress, arguing that the SKN-1t transgenes recapitulate the transcriptional outputs of endogenous SKN-1 isoforms (Fig 2 and S3 and S4 Tables).

Sequence editing alters the transcriptional output of activated SKN-1t in a manner that clarifies the functional distinction between SKN-1A and SKN-1C. On the one hand, extensive sequence editing is required for SKN-1t transgenes to drive upregulation of proteasome subunit genes and other factors linked to proteasome assembly or function (Fig 2B–2D and S3 Table). This observation matches the essential role of sequence edited SKN-1A and Nrf1 in proteasome regulation and proteostasis [21,22]. On the other hand, extensive sequence editing dampens (but does not eliminate) SKN-1t transgenic animals' activation of many genes that are strongly activated by non-edited SKN-1t. Genes in this category are strongly associated with xenobiotic detoxification and oxidative stress response functions (Fig 2B–2I and S4 Table). Importantly, this overlap in transcriptional control is conserved and functionally significant, as SKN-1A/Nrf1 and SKN-1C/Nrf2 act in concert to ensure optimal oxidative stress defenses (discussed further below). In sum, sequence editing sculpts gene induction by SKN-1A/Nrf1 to allow its unique function in regulation of proteasome biogenesis but also influences its functional overlap with SKN-1C/Nrf2.

To undergo sequence editing, SKN-1A/Nrf1 must first be N-glycosylated in the ER. The patterns and overall levels of site-specific N-glycosylation, along with the structures of the attached glycans, differ between cell types, are remodeled in response to stress, and are mis-regulated in disease [80–83]. In certain conditions, including congenital disorders of glycosylation or diseases linked to ER stress, sites that are normally uniformly N-glycosylated, can show reduced N-glycan occupancy [82,84]. These observations raise the possibility that under some physiological, stress, or disease states, a percentage of SKN-1A/Nrf1 may exit the ER in a partially N-glycosylated state. Our findings suggest that variation in the extent of SKN-1A/Nrf1 N-glycosylation, and the consequent changes in sequence editing, may impact its ability to activate proteasome biogenesis and/or control redox homeostasis, therefore altering cellular stress responses. We speculate that the extent of SKN-1A/Nrf1 N-glycosylation itself might be regulated, providing a mechanism to fine-tune stress responses in a cell type- or context-specific manner. Modulating Nrf1 function via fine-tuning N-glycosylation could also be exploited therapeutically, as blocking Nrf1 processing has been proposed as a strategy to enhance the efficacy of proteasome inhibitor drugs, such as BTZ, in cancer treatment [18,29]. Our findings suggest that pharmacological interventions that reduce Nrf1 N-glycosylation, thereby limiting the extent of sequence editing, may be beneficial in this context. To explore the physiological and potential disease relevance of partial sequence editing, it will be crucial to precisely measure the extent of N-linked glycosylation of SKN-1A/Nrf1 in diverse cell types and in different physiological and disease states. Although biochemical analysis of Nrf1 processing is technically challenging [23], one immunopeptidomic study detected both deamidated Nrf1 peptides, that are presumed to be produced via deglycosylation, as well as non-deamidated species, suggesting that partial sequence editing of Nrf1 is possible [85].

Many SKN-1t-upregulated genes contain SKN-1 binding sites in their promoters and are likely to be directly regulated [86,87]. The residues that are affected by sequence editing are within a putative transactivation domain and are not likely to directly alter the sequence specificity of DNA binding [86,88]. Intriguingly, mutations that affect different subsets of Nrf1 glycosylation sites cause distinct effects on transactivation of a reporter transgene, suggesting that different patterns of sequence editing have distinct effects on the ability of Nrf1 to activate gene transcription [89]. Sequence editing patterns may control interactions between SKN-1A/Nrf1 and transcriptional co-activators or other factors that regulate SKN-1A/Nrf1 function. Identification of these factors and understanding how their interactions with SKN-1A/Nrf1 are affected by sequence editing will be an important area for future investigation.

**SKN-1A/Nrf1 and SKN-1C/Nrf2 act in concert to promote oxidative stress resistance**

We demonstrate that both SKN-1A and SKN-1C contribute to optimal oxidative stress defenses. Our genetic analysis supports a model in which SKN-1C acts downstream of a signaling cascade involving XREP-4 and WDR-23. However, SKN-1A is not regulated by the XREP-4/WDR-23 signaling cascade, so its participation in oxidative stress responses must be governed by other regulatory mechanisms. SKN-1A is typically degraded by the proteasome. Exposure to AS or other agents that cause oxidative stress may lead to impaired proteasome function, leading to SKN-1A stabilization and enabling its transcriptional function [90,91]. Although further study is needed to fully understand the mechanism(s) governing SKN-1A activity under oxidative stress, our genetic analysis indicates that deglycosylation by PNG-1 is required. Thus, although they are regulated by distinct signaling pathways, sequence edited SKN-1A acts in concert with SKN-1C to promote oxidative stress resistance, reflecting their overlapping effects on gene expression. Many genes with functions related to detoxification and redox homeostasis can be regulated by both SKN-1C and sequence edited SKN-1A, whereas genes related to proteasome biogenesis and function are solely controlled by sequence edited SKN-1A. The overlapping functions of SKN-1A and SKN-1C in transcriptional control are conserved with mammalian Nrf1 and Nrf2 respectively. Sequence editing of Nrf1 is required for regulation of proteasome subunit genes expression, which is not controlled by Nrf2 [24]. Genes with redox-related functions are regulated by both Nrf1 and Nrf2 [47,50,51,92].

Sequence edited SKN-1A may promote oxidative stress resistance in part by bolstering proteasome biogenesis. Oxidative stress can cause defects in proteasome assembly or function [91]. In addition, the proteasome contributes to cellular survival under oxidative stress by degrading oxidatively damaged proteins and peptides [93]. However, transcriptomic analyses indicate that oxidative stress does not always trigger proteasome upregulation in *C. elegans* [57,58,77]. Similarly, transcriptional reporters of proteasome subunit gene expression do not always show upregulation under oxidative stress conditions [21,94,95]. These data suggest that sequence-edited SKN-1A can contribute to oxidative stress defense without boosting proteasome levels, instead regulating genes that are also induced by SKN-1C. In these cases, induction might primarily depend on SKN-1C, and SKN-1A may modulate the timing or magnitude of induction, as is the case for the *gst-4*$_p$*::gfp* reporter (Fig 6E–6L). It is interesting that the relative contributions of SKN-1A and SKN-1C to growth or viability differs under different oxidative stress conditions (Fig 6A–6D). We therefore suggest that coordinated control of partly overlapping transcriptional outputs by SKN-1A/Nrf1 and SKN-1C/Nrf2 allows flexibility to tailor appropriate responses to different forms of oxidative stress or to stress experienced at different stages of development or in different cell types.

**Coordination of SKN-1A/C functions promotes organismal homeostasis**

Loss of *skn-1* leads to defects in many aspects of organismal and cellular homeostasis even in the absence of exogenous proteotoxic or oxidative stress [11,34]. Hyperactivation of SKN-1, although it can render animals highly stress resistant, can also have profound detrimental effects [60,96–101]. Thus, *skn-1* controls gene expression to ensure homeostasis as well as to induce stress responses. Given that distinct pathways regulate SKN-1A and SKN-1C, precise homeostatic control of gene expression must be achieved by coordinated control of both isoforms. Interestingly, phosphorylation, O-GlcNAcylation, and arginine methylation are all implicated in regulation of SKN-1 function, but whether these modifications occur or exert their effects in an isoform-specific manner is not fully resolved [33,67,73,94,102,103]. Conceivably, these modifications could affect both SKN-1A and SKN-1C, occur in an isoform-specific manner, or have differential effects on each isoform's function. Future investigation of the impact of these modifications on each SKN-1 isoform, and how they may intersect with the effects of sequence editing, is needed to fully understand their roles in *skn-1*-dependent stress-responses and homeostasis.

Transcriptional autoregulation and/or cross-regulation may be an additional mechanism that precisely controls the activities of SKN-1A and SKN-1C across different physiological contexts. WDR-23, a critical regulator of SKN-1C, is transcriptionally upregulated by all three SKN-1t transgenes (~5.5-fold by SKN-1t[NNNN], ~4-fold b SKN-1t[NDNN], ~2.5-fold by SKN-1t[DNDD]). Elevation of WDR-23 levels by SKN-1A/C may limit SKN-1C activation through a negative feedback loop.

In contrast, our analysis of the *skn-1a* and *skn-1c* promoters suggests that *skn-1a* can be transcriptionally upregulated following activation of either SKN-1A or SKN-1C, suggesting a positive feedback loop. These two isoform-specific autoregulatory mechanisms may augment SKN-1A levels under certain conditions while dampening SKN-1C activity. Interestingly, transcriptional activation of the *skn-1a* reporter appears to occur solely in the intestine, suggesting that isoform-specific autoregulatory feedback mechanisms may function differently in different tissues.

### SKN-1A/C functions and longevity

Genetic analysis in *C. elegans* indicates a crucial role for *skn-1* in longevity. Inactivation of *skn-1* (by mutations that inactivate both SKN-1A and SKN-1C) abrogates the effects of genetic or environmental interventions that extend lifespan including the conserved insulin/IGF and mTOR signaling pathways [67,68,70,104]. Isoform specific analysis of SKN-1A and SKN-1C functions provide the basis to investigate how each isoform contributes to different genetic or environmental interventions that extend lifespan. Given the conservation of their mechanisms of regulation and functions, isoform-specific effects of SKN-1A and SKN-1C on lifespan in *C. elegans* may point to potential roles of Nrf1 and Nrf2 in human longevity and age-related disease.

## Materials and methods

### *C. elegans* maintenance and genetics

*C. elegans* was maintained at 20°C on standard nematode growth media (NGM) and fed *E. coli* OP50, unless otherwise noted. A list of strains used in this study is provided in S6 Table. Some strains were provided by the CGC, which is funded by the NIH Office of Research Infrastructure Programs (P40 OD010440). *png-1(ok1654)* was generated by the *C. elegans* Gene Knockout Project at the Oklahoma Medical Research Foundation, part of the International *C. elegans* Gene Knockout Consortium.

### RNAseq sample preparation and sequencing

Animals were bleach-synchronized and hatched overnight in M9 at 20°C while rotating. Approximately 2,000 synchronized L1 larvae were plated onto 9 cm NGM plates and grown to the L4 stage. Animals were then washed 3x with M9 and the worm pellet was resuspended in 1mL of TRIzol reagent (Invitrogen) before being stored at -80°C. Four replicate samples were collected for each genotype and RNA was extracted according to manufacturer's instructions. Total RNA integrity was assessed using an Agilent 4200 TapeStation (Agilent Technologies, Inc.) and quantified using a Trinean DropSense96 spectrophotometer (Caliper Life Sciences). RNA-seq libraries were prepared from total RNA using the TruSeq Stranded mRNA kit (Illumina, Inc.). Library size distribution was validated using an Agilent 4200 TapeStation (Agilent Technologies). Additional library QC, blending of pooled indexed libraries, and cluster optimization was performed using Life Technologies' Invitrogen Qubit 2.0 Fluorometer (Life Technologies-Invitrogen). RNA-seq libraries were pooled (16-plex) and clustered onto a P2 flow cell. Sequencing was performed using an Illumina NextSeq 2000 employing a paired-end, 50-base read length (PE50) sequencing strategy.

### RNAseq analysis: Differentially expressed genes

STAR v2.7.7a [105] with 2-pass mapping was used to align paired-end reads to WBcel235 genome assembly and quantify gene-level counts based on ENSEMBL gene annotation v52. FastQC 0.11.9 (https://www.bioinformatics.babraham.ac.uk/projects/fastqc/). RNA-SeQC 2.3.4 [106] was used to check read QC metrics. Bioconductor package edgeR (3.26.8) [107] was used to detect differential gene expression between sample groups. Genes with low expression were excluded using the edgeR function filterByExpr with min.count = 10 and min.total.count = 15. The filtered expression matrix was normalized using the TMM method [108] and subjected to significance testing using the quasi-likelihood pipeline implemented

in edgeR. In the primary analysis, a gene was considered differentially expressed if the absolute log2 fold change was greater than 1 (i.e., fold change > 2 in either direction), and the Benjamini-Hochberg adjusted p-value below 0.01. In the secondary analysis, we considered a gene to be differentially expressed (with lower stringency) if the absolute log2 fold change was greater than 0.585 (i.e., fold change > 1.5 in either direction) and the Benjamini-Hochberg adjusted p-value was below 0.05. RNAseq data and analysis are deposited under GEO accession GSE279082.

### RNAseq analysis: Gene set enrichment

We divided SKN-1t-activated genes into three classes according to the effect of sequence-editing-mimetic mutations. The classes are: (1) 'high-D'. Genes included in this set meet the following criteria: upregulated (>2-fold, FDR < 0.01) by SKN-1t[NDNN], NOT SKN-1t[NNNN]; (2) 'overlap'. Genes included in this set meet the following criteria: upregulated (>2-fold, FDR < 0.01) by SKN-1t[NNNN], SKN-1t[NDNN] and SKN-1t[DNDD]; (3) 'low-D'. Genes included in this set meet the following criteria: upregulated (>2-fold, FDR < 0.01) by SKN-1t[NNNN] OR SKN-1t[NDNN], NOT SKN-1t[DNDD]).

Enrichment analysis was carried out for the genes upregulated by each SKN-1t transgene as well as for the three classes described above. Functional enrichment analysis was carried out using WormCat [53]. Published lists of stress-responsive and *skn-1*-regulated genes were processed using the Gene Name Sanitizer tool on WormBase [109]. Enrichment factors and the significance of overlap between upregulated gene sets were calculated using a hypergeometric test (nemates.org). The most conservative estimate of genome size (14,022), corresponding to the number of genes detected in our RNA-seq dataset was used for these tests.

### Genome modification by CRISPR/Cas9

CRISPR/Cas9 genome editing was performed by microinjection of guideRNA/Cas9 RNPs (IDT #1081058, #1072532, and custom cRNA) and ssDNA oligos (IDT) as homology-directed repair templates [110,111]. The *skn-1*[M1A/M91A] edit was generated with a guide RNA targeting the sequence: TGTACACGGACAGCAATAAT, and the following ssDNA repair template: tatacaaaactatgatatatatttcagAAgctTAtACcGAttctAAcAAcAGaAACTTTGATGAAGTCAACCATCAGCATCAA. The edit replaces the start codon of the *skn-1c* open reading frame with an alanine and creates a silent mutation in the adjacent codon which destroys an RsaI restriction site to facilitate genotyping. The *xrep-4*[E22STOP] edit was generated with a guide targeting the sequence: ATATTTCTGCGTATTTCTCG, and the following ssDNA repair template: GAGGCTGGAT-GGAAGCACTTGCCACGAtgatcatAAATATTCTCACTCGACTACCATTCC. The edit replaces the 22nd codon of the *xrep-4* coding sequence with a premature termination codon, alters the reading frame, and introduces a BclI restriction site to facilitate genotyping. Desired edits were detected by diagnostic PCR and subsequently confirmed by sequencing.

### Plasmid constructs and transgenesis

Cloning was carried out using the NEBuilder HiFi DNA Assembly kit (New England Biolabs #E2621). All plasmids were assembled into pNL43, a modified version of pCFJ909 containing the pBluescript MCS [20]. Integrated transgenes were generated using CRISPR/Cas9 to direct insertion of multicopy arrays at selected safe-harbor insertion sites [112,113]. Details of each construct and transgenics generated are described below.

*skn-1a$_p$::mCherry::H2B (pNL514).* We cloned the CEOP4172 promoter (3019 bp immediately upstream of the *bec-1* start codon) upstream of the mCherry::H2B coding sequence (mCherry fused in-frame to the *his-58* coding sequence), and the *tbb-2* 3'UTR (376 bp immediately downstream of the *tbb-2* stop codon). A multicopy array containing pNL514 and carrier DNA (Invitrogen 1 kb plus DNA ladder, #10787018) was integrated into *nicTi601* on chromosome II.

*skn-1c$_p$::mCherry::H2B (pNL515).* We cloned the *skn-1c* promoter (4159 bp immediately upstream of the *skn-1c* start codon) upstream the mCherry::H2B coding sequence (mCherry fused in-frame to the *his-58* coding sequence), and the *tbb-2* 3'UTR (376 bp immediately downstream of the *tbb-2* stop codon). A multicopy array containing pNL515 and carrier DNA (Invitrogen 1 kb plus DNA ladder, #10787018) was integrated into *nicTi601* on chromosome II.

***gst-4<sub>p</sub>::mCherry* transcriptional reporter (pNL490).** We cloned the *gst-4* promoter (727 bp immediately upstream of the *gst-4* start codon) upstream of the mCherry coding sequence, and the *tbb-2* 3'UTR (376 bp immediately downstream of the *tbb-2* stop codon). A multicopy array containing pNL490 and carrier DNA (Invitrogen 1 kb plus DNA ladder, #10787018) was integrated into *nicTi602* on chromosome V.

## Bortezomib resistance assays

Bortezomib sensitivity was assessed by the ability of animals to develop, or survive as adults, on plates supplemented with bortezomib at various concentrations (LC Laboratories, #B1408). The bortezomib solution was directly applied to NGM plates seeded with OP50 and allowed to dry for 2h under the hood. Control plates supplemented with DMSO were prepared in parallel.

For development assays, 5 L4-stage animals were transferred to fresh plates supplemented with 0.04 µg/ml (104 nM) bortezomib and incubated at 20°C for three or four days. Progeny development was then assessed. For mutants balanced with tmC25 (which contains a *myo-2<sub>p</sub>*::GFP marker), homozygous mutant animals were separated by selecting animals that did not express the marker to fresh bortezomib-supplemented or control plates. Images were captured using a Leica M165FC microscope equipped with a Leica K5 sCMOS camera and LAS X software.

For survival assays, 30 L4 stage animals were picked to fresh plates supplemented with 0.4 µg/ml (1.04 µM) bortezomib and the animals' survival was scored after 4 days. For strains that produce progeny that can develop in the presence of the tested concentration of bortezomib, animals were transferred to fresh bortezomib-supplemented plates after 2–3 days to ensure that the animals being assayed for survival could be distinguished from their progeny.

## Oxidative stress resistance assays

**Arsenite survival assay.** NGM plates supplemented with sodium arsenite (RICCA Chemical; Cat # 7142-16) at final concentrations of 1 mM, 2 mM, or 3 mM were prepared the day prior to the assay. Plates were left to dry for 2 hours before being seeded with 250 µL of 10X concentrated OP50. Approximately 30 L4-stage *C. elegans* were transferred to each plate the following day and incubated at 20°C for 24 hours. After incubation, the number of live and dead animals was recorded.

**Paraquat development assay.** NGM plates supplemented with paraquat were prepared the day prior to the experiment. A 1 M stock of methyl viologen hydrate (Thermo Scientific; Cat # 227320050) in ddH₂O was freshly prepared and added to the NGM media at a final concentration of 4 mM. Plates were left to dry for 2 hours and then seeded with 250 µL of 10X concentrated OP50. 10–25 gravid animals were transferred to each plate the following day and incubated at 20°C for 3–4 hours. After incubation, the adults were removed, and the progeny were allowed to develop for 7 days. The number of animals that developed into gravid adults was recorded, along with the total number of animals on the plate. For mutants balanced with tmC25, only homozygous mutant progeny (identified by the absence of the *myo-2<sub>p</sub>*::GFP marker) were scored.

## Preparation of 10X OP50 for oxidative stress assays

A single colony of OP50 was inoculated in 5 mL of Luria-Bertani (LB) broth and grown over night at 37°C. The following day, 1 mL of this culture was inoculated into 1 L of Terrific Broth (TB) and incubated at 37°C for 2 days. The culture was then centrifuged at 4,000 rpm for 15 min. The supernatant was discarded, and the pellet was resuspended in 45 mL of M9 and transferred to a 50 mL conical tube, creating a 20X stock solution (stored at 4°C). The 20X stock was subsequently diluted 1:1 with M9 to create a 10X working solution.

## Embryonic lethality

For each assayed genotype, 10 gravid adult animals (selected at random from mixed-stage plates containing day 1 and day 2 adults) were moved to a fresh plate and allowed to lay eggs for ~4 hours. 50 of the newly laid eggs were then transferred to a fresh plate. The number of unhatched eggs after 24 hours was recorded.

## Lifespan measurements

For lifespan assays, approximately 70 L4-stage animals were randomly selected from mixed-stage cultures that had been maintained without starvation for at least two generations and transferred onto NGM plates seeded with OP50. Animals were transferred to fresh plates on day three and then every 2 days, until reproduction ceased, and every 3–5 days thereafter. Animals were checked for survival at least every other day. Animals that died by bagging or crawling off the plates were censored, while those that died due to age-related vulval integrity defects (after ceasing reproduction, when such defects can be distinguished from bagging) were not censored. Survival curves, calculation of mean lifespan, and statistical analysis was performed using Graphpad Prism. The Log-rank (Mantel-Cox) test was used to compare survival curves. Raw data and statistics for all lifespan assays are shown in S5 Table.

## Adult vulval integrity defect (Avid) assay

L4 animals were selected at random from mixed stage cultures that had been maintained without starvation for at least two generations and transferred onto NGM plates seeded with OP50. Animals were transferred to fresh plates on day 3 and day 5 of adulthood. On day 5–7, animals were checked for rupture, and the cumulative total number of animals that ruptured was recorded. At least 50 animals were analyzed in each replicate assay.

## Drug/stress treatments for reporter imaging

**Oxidative stress.** NGM plates containing arsenite (2 or 3 mM) or paraquat (3 mM) were prepared the day before the imaging assay as described for the oxidative stress resistance assays. For juglone induction, a juglone (Cayman Chemical; Cat # 481-39-0) solution was directly applied to NGM plates seeded with OP50, resulting in a final concentration of 38 µM, and allowed to dry for 2 hours under the hood. Control plates supplemented with DMSO were prepared in parallel. Approximately 30 L4-stage *C. elegans* were transferred to each plate and incubated at 20°C for 4–20 hours. Animals were then imaged using the procedure described in the Microscopy section.

**Proteasome inhibition.** NGM plates containing 0.4 µg/ml (1.04 µM) bortezomib were prepared as described for bortezomib resistance assays. Control plates supplemented with DMSO were prepared in parallel. Approximately 30 L4-stage *C. elegans* were transferred to each plate and incubated at 20°C for approximately 24 hours. For imaging of Ub[G76V]::GFP accumulation following bortezomib treatment, animals were raised to the L4 stage at 25°C before transfer to bortezomib-supplemented plates. Ub[G76V]::GFP does not accumulate to detectable levels in any of the mutants if they are raised to the L4 stage at 25°C (see S5 Fig). Thus, challenging L4-stage animals raised at 25°C ensures that Ub[G76V]::GFP accumulation only occurs after drug treatment. Animals were imaged using the procedure described in the Microscopy section.

## Microscopy

Brightfield and fluorescence images were collected on a Leica M165FC equipped with a Leica K5 sCMOS camera and using LAS X software. For fluorescence imaging, worms were immobilized using sodium azide and mounted on 2% agarose pads. For all fluorescence images, images shown within the same figure panel were collected using the same exposure time and were then processed identically. To quantify *rpt-3$_p$::gfp*, *gst-4$_p$::gfp*, and *gst-4$_p$::mcherry* expression, the mean pixel intensity along a longitudinal section of each animal was measured. To quantify *rpl-28$_p$::ub[G76V]::GFP* accumulation in L4 animals (without bortezomib treatment), mean pixel intensity in the two anterior-most intestinal cells was measured. To quantify *rpl-28$_p$::ub[G76V]::GFP* accumulation in adult animals following bortezomib or DMSO control treatment, the mean pixel intensity along a longitudinal section of each animal was measured. All Image processing and analysis was performed using Fiji software [114].

## Statistical analysis

Statistical analysis for RNAseq and lifespan experiments are each described in the relevant section of the methods. All other statistical analyses were performed using Graphpad Prism. All biological replicates were performed using independent populations of animals.

## Supporting information

**S1 Fig. Differential gene expression in SKN-1t transgenic strains.** A-C) Volcano plots showing differential gene expression in SKN-1t transgenic strains compared to the non-transgenic wild-type control. Significantly upregulated and downregulated genes (fold change >2, FDR<0.01) are indicated in red and in blue, respectively. Genes that do not show significant differential expression are indicated in black. D) Principal Component Analysis comparison of gene expression profiles showing non-transgenic control and SKN-1t transgenic samples.
(TIF)

**S2 Fig. Most genes that appear to be uniquely upregulated in a single SKN-1t transgenic strain are upregulated in one or more additional SKN-1t transgenic strain.** A) Analysis of 189 genes that appear to be uniquely upregulated in SKN-1t[NNNN] transgenic animals (>2-fold, FDR<0.01). (i, ii) Volcano plots showing differential expression of uniquely SKN-1t[NNNN]-upregulated genes in SKN-1t[NDNN] (i) and SKN-1t[DNDD] (ii) transgenic animals. In each graph, the 189 genes are indicated in black, all other genes are indicated in gray. (iii) Violin plot comparing log2 fold change of the 189 uniquely SKN-1t[NNNN]-upregulated genes in each SKN-1t transgenic strain. These genes are skewed towards upregulation in SKN-1t[NDNN] and SKN-1t[DNDD] transgenic animals. (iv) Proportion of the 189 uniquely SKN-1t[NNNN]-upregulated genes that are upregulated in SKN-1t[NDNN] and/or SKN-1t[DNDD] at a lower stringency cutoff (>1.5-fold, FDR<0.05). Most genes are upregulated in SKN-1t[NDNN] transgenics, and a smaller fraction are also upregulated in SKN-1t[DNDD] transgenics at this cutoff. B) Analysis of 99 genes that appear to be uniquely upregulated in SKN-1t[NDNN] transgenic animals (>2-fold, FDR<0.01). (i, ii) Volcano plots showing differential expression of uniquely SKN-1t[NDNN]-upregulated genes in SKN-1t[NNNN] (i) and SKN-1t[DNDD] (ii) transgenic animals. In each graph, the 99 genes are indicated in black, all other genes are indicated in gray. (iii) Violin plot comparing log2 fold change of the 99 uniquely SKN-1t[NDNN]-upregulated genes in each SKN-1t transgenic strain. These genes are skewed towards upregulation in SKN-1t[NNNN] and SKN-1t[DNDD] transgenic animals. (iv) Proportion of the 99 uniquely SKN-1t[NNNN]-upregulated genes that are upregulated in SKN-1t[NDNN] and/or SKN-1t[DNDD] at a lower stringency cutoff (>1.5-fold, FDR<0.05). Most genes are upregulated in at least one of the two other transgenic strains at this cutoff. C) Analysis of 125 genes that appear to be uniquely upregulated in SKN-1t[DNDD] transgenic animals (>2-fold, FDR<0.01). (i, ii) Volcano plots showing differential expression of uniquely SKN-1t[DNDD]-upregulated genes in SKN-1t[NNNN] (i) and SKN-1t[NDNN] (ii) transgenic animals. In each graph, the 125 genes are indicated in black, all other genes are indicated in gray. (iii) Violin plot comparing log2 fold change of the 125 uniquely SKN-1t[DNDD]-upregulated genes in each SKN-1t transgenic strain. These genes are skewed towards upregulation in SKN-1t[NDNN] but not SKN-1t[NNNN] animals. (iv) Proportion of the 125 uniquely SKN-1t[DNDD]-upregulated genes that are upregulated in SKN-1t[NNNN] and/or SKN-1t[NDNN] transgenics at a lower stringency cutoff (>1.5-fold, FDR<0.05). Most genes are upregulated in SKN-1t[NDNN] transgenics at this cutoff and a smaller fraction are also activated in SKN-1t[NNNN] transgenics.
(TIF)

**S3 Fig. Some genes that appear to be upregulated only in two of the three SKN-1t transgenic strains are additionally upregulated in the third SKN-1t transgenic strain.** A) Analysis of 517 genes that appear to be upregulated in SKN-1t[NNNN] and SKN-1-NDNN] transgenic animals (>2-fold, FDR<0.01), but not in SKN-1t[DNDD] transgenic animals. (i) Volcano plots showing differential expression of these 517 genes SKN-1t[DNDD] transgenic animals. The 517 genes

are indicated in black; all other genes are indicated in gray. (ii) Proportion of these 517 genes that are upregulated in SKN-1t[DNDD] transgenics at a lower stringency cutoff (>1.5-fold, FDR<0.05). Most genes are upregulated in SKN-1t[DNDD] transgenics at this cutoff. B) Analysis of 74 genes that appear to be upregulated in SKN-1t[NDNN] and SKN-1-DNDD] transgenic animals (>2-fold, FDR<0.01), but not in SKN-1t[NNNN] transgenic animals. (i) Volcano plots showing differential expression of these 74 genes SKN-1t[NNNN] transgenic animals. The 74 genes are indicated in black; all other genes are indicated in gray. (ii) Proportion of these 75 genes that are upregulated in SKN-1t[NNNN] transgenics at a lower stringency cutoff (>1.5-fold, FDR<0.05). Approximately one third of the 74 genes are upregulated in SKN-1t[NNNN] transgenics at this cutoff.
(TIF)

**S4 Fig. Functional enrichments of SKN-1t-upregulated genes.** Functional enrichments of genes upregulated in each SKN-1t transgenic strain (see Fig 1A and 1B), as assessed by WormCat.
(TIF)

**S5 Fig. Functional enrichments of SKN-1t-upregulated genes, classified according to the effect of sequence editing.** Functional enrichments of SKN-1t-upregulated genes categories that are differentially affected by sequence editing mutations, as assessed by WormCat. The three classes are defined as follows (see Fig 2A for more detail): (1) 'high-D' genes are upregulated (>2-fold, FDR<0.01) in SKN-1t[DNDD], but not in SKN-1t[NNNN] transgenics; (2) 'overlap' genes are upregulated (>2-fold, FDR<0.01) in SKN-1t[NNNN], SKN-1t[NDNN] and SKN-1t[DNDD] transgenics; (3) 'low-D' genes are upregulated in SKN-1t[NNNN] and/or SKN-1t[NDNN], but not in SKN-1t[DNDD] transgenics.
(TIF)

**S6 Fig. Effect of *skn-1* mutations of proteasomal degradation of Ub[G76V]::GFP.** A) Fluorescence images showing accumulation of Ub[G76V]::GFP in *skn-1* mutants at the L4 stage. Increased accumulation is observed in *skn-1a* and *skn-1ac* mutants, but not in *skn-1c* mutants. Scale bar shows 100 µm. B) Quantification of Ub[G76V]::GFP in *skn-1* mutants raised at different temperatures to the L4 stage. At 15°C and 20°C, increased Ub[G76V]::GFP is detected in *skn-1a* and *skn-1ac* mutants, but not *skn-1c* mutant animals. In animals raised at 25°C, there are no detectable changes in any of the mutants. n=30 animals were imaged for each genotype at each temperature. Error bars show mean±SD. **** $p<0.0001$, *** $p<0.001$, ordinary two-way ANOVA with Tukey's multiple comparisons test. Numerical data for panels B-D are available in S1 Data.
(TIF)

**S7 Fig. Transcriptional reporters for *skn-1a* and *skn-1c* show ubiquitous expression.** A) Schematic showing CEOP4172 including the *bec-1* and *skn-1* loci. The DNA fragments corresponding to the promoter of *skn-1a*/COEP4172 (*skn-1a*$_p$) and the promoter of *skn-1c* (*skn-1c*$_p$) are shown. B) Fluorescence image showing expression of mCherry::H2B under the *skn-1a*/CEOP4172 promoter at each stage of larval development and in adults. The reporter is expressed at all stages and in most cells. The most prominent expression is detected in unidentified cells in the head. Scale bar shows 100 µm. C) Fluorescence image showing expression of mCherry::H2B under the *skn-1c* promoter at each stage of larval development and in adults. The reporter is expressed at all stages in most cells. The most prominent expression is detected in the intestine. Scale bar shows 100 µm. D, E) Fluorescence micrographs showing skn-1a$_p$::mCherry::H2B (d) and skn-1c$_p$::mCherry::H2B (e) expression in animals exposed to the proteasome inhibitor bortezomib (BTZ, 0.4 µg/ml (1.04 µM)) or Arsenite (3mM) compared to untreated controls. In each case, expression of the reporter is unchanged under stress conditions. Scale bar shows 100 µm. F, G) Fluorescence micrographs showing *skn-1a*$_p$::mCherry::H2B (F) and *skn-1c*$_p$::mCherry::H2B (G) expression in *wdr-23*(*tm1817*) mutants and in SKN-1t[DNDD] transgenics compared to wild type controls. Images show L4 stage animals. The expression of the *skn-1c* reporter is unchanged, whereas the expression of the *skn-1a* reporter is increased in intestinal cells. Scale bar shows 100 µm. H) Fluorescence micrographs

showing increased expression of *skn-1a_p_::mCherry::H2B* in intestinal cells of *wdr-23*(*tm1817*) mutant animals and SKN-1t[DNDD] transgenic animals compared to wild type controls. Images show L4 stage animals. Scale bar shows 20 µm.
(TIF)

**S8 Fig. Inactivation of *xrep-4* disrupts regulation of *gst-4_p_::gfp* but not the proteasome subunit reporter *rpt-3_p_::gfp*.** A) Fluorescence micrographs showing *gst-4_p_::gfp* expression in animals exposed to juglone (38 µM, 4 hours) or paraquat (3 mM, 4 hours). The reporter is induced in the wild type but not in *xrep-4* mutant animals. Scale bar shows 100 µm. B, C) Quantification of *gst-4_p_::gfp* induction in animals exposed to juglone (38 µM, 4 hours) or paraquat (3 mM, 4 hours), as shown in (A). Error bars show mean ± SD. **** $p < 0.0001$, ns $p > 0.05$, ordinary two-way ANOVA with Sidak's multiple comparisons test. D) Fluorescence micrographs showing *gst-4_p_::gfp* expression in animals exposed to arsenite (3 mM, 4 hours). Induction of the reporter is attenuated in *xrep-4* mutant animals. Scale bar shows 100 µm. E) Quantification of *gst-4_p_::gfp* expression in animals exposed to arsenite (3 mM, 4 hours), as shown in (D). Error bars show mean ± SD. **** $p < 0.0001$, ordinary two-way ANOVA with Sidak's multiple comparisons test. F) Fluorescence micrographs showing *rpt-3_p_::gfp* expression in animals exposed to bortezomib (BTZ, 0.4 µg/ml, approximately 20 hours). Induction is not altered in *xrep-4* mutants compared to the wild type. Scale bar shows 100 µm. G) Quantification of *rpt-3_p_::gfp* expression in animals exposed to bortezomib (BTZ, 0.4 µg/ml), as shown in (F). Error bars show mean ± SD. **** $p < 0.0001$, ordinary two-way ANOVA with Sidak's multiple comparisons test. Numerical data for panels B, C, E, and G are available in S1 Data.
(TIF)

**S1 Table. Differentially expressed genes in SKN-1t transgenic strains.**
(XLSX)

**S2 Table. Categories of SKN-1t-upregulated genes that are differentially sensitive to sequence editing.**
(XLSX)

**S3 Table. Genes from the 'high-D' category that are upregulated in animals exposed to bortezomib.**
(XLSX)

**S4 Table. Genes from the 'overlap' category that are upregulated in animals exposed to oxidative stress.**
(XLSX)

**S5 Table. Raw data from all lifespan assays.**
(XLSX)

**S6 Table. *C. elegans* strains used in this study.**
(XLSX)

**S1 Data. Numerical data for all graphs presented in this study.**
(XLSX)

## Acknowledgments

We thank Wormbase for *C. elegans* genome database curation and data access. Some strains were provided by the CGC, which is funded by NIH Office of Research Infrastructure Programs (P40 OD010440).

## Author contributions

**Conceptualization:** Nicolas Lehrbach.

**Funding acquisition:** Nicolas Lehrbach.

**Investigation:** Briar Jochim, Irini Topalidou.

**Methodology:** Briar Jochim, Irini Topalidou, Nicolas Lehrbach.

**Supervision:** Nicolas Lehrbach.

**Visualization:** Briar Jochim, Irini Topalidou, Nicolas Lehrbach.

**Writing – original draft:** Nicolas Lehrbach.

**Writing – review & editing:** Irini Topalidou, Nicolas Lehrbach.

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
