## [Decision Letter · Decision Letter 0]

PGENETICS-D-25-00366

Protein sequence editing defines distinct and overlapping functions of SKN-1A/Nrf1 and SKN-1C/Nrf2.

PLOS Genetics

Dear Dr. Lehrbach,

Thank you for submitting your manuscript to PLOS Genetics. After careful consideration, we feel that it has merit but does not fully meet PLOS Genetics's publication criteria as it currently stands. Therefore, we invite you to submit a revised version of the manuscript that addresses the points raised during the review process.

Please submit your revised manuscript within 30 days May 31 2025 11:59PM. If you will need more time than this to complete your revisions, please reply to this message or contact the journal office at plosgenetics@plos.org. Please include the following items when submitting your revised manuscript:

We look forward to receiving your revised manuscript.

Kind regards,

Laura Bianchi

Academic Editor

PLOS Genetics

Marnie Blewitt

Section Editor

PLOS Genetics

Aimée Dudley

Editor-in-Chief

PLOS Genetics

Anne Goriely

Editor-in-Chief

PLOS Genetics

**Journal Requirements:**

https://journals.plos.org/plosgenetics/s/submission-guidelines#loc-parts-of-a-submission

- ® on page: 23.

5) We have noticed that you have uploaded Supporting Information files, but you have not included a list of legends. Please add a full list of legends for your Supporting Information files after the references list.

6) Please ensure that the funders and grant numbers match between the Financial Disclosure field and the Funding Information tab in your submission form. Note that the funders must be provided in the same order in both places as well. State what role the funders took in the study. If the funders had no role in your study, please state: "The funders had no role in study design, data collection and analysis, decision to publish, or preparation of the manuscript.".

**Reviewers' comments:**

Reviewer's Responses to Questions

**Comments to the Authors:**

Reviewer #1: In this manuscript, Jochim et al. elucidate the functional distinctions and cooperative interactions between the two major isoforms of the transcription factor SKN-1, namely SKN-1A and SKN-1C, in C. elegans. Through the use of a constitutively active SKN-1t transgenes, the authors demonstrate that protein sequence editing in SKN-1A is essential for the activation of proteasome genes while simultaneously dampening the activation of oxidative stress response genes. Although previous work from the authors' group has established that PNG-mediated editing of SKN-1A is critical for proteasome gene induction, this study provides novel insights by showing that the edited SKN-1A can also induce stress response factors. Furthermore, comparative analyses using mutant alleles reveal that SKN-1A and SKN-1C function cooperatively to enhance oxidative stress resistance, highlighting distinct but complementary regulatory mechanisms. These findings significantly advance our understanding of SKN-1 isoform regulation in stress responses and have important implications for disease and longevity.

The following recommendations are provided to assist the authors:

Major

• The authors constructed three SKN-1t to investigate the effects of different degrees of sequence editing on transcriptional outputs. In this study, NDNN and DNDD are used as the edited variants. However, it remains unclear which of the sequence editing at four glycosylation sites in SKN-1A are most critical for proteasome gene transcription, or whether all four are equally important. For example, if editing at N403 is crucial, the weak induction activity of NDNN would be reasonable. Conversely, if N338 is more important, DNDD may exhibit less transcriptional activation. Since the edited sites differ completely between variants, a simple numerical comparison is not feasible. It would therefore be preferable if only one D mutant aligned with the DNDD variant. Is there evidence that editing at a single site yields transcriptional activity identical to that of the fully edited form? An explanation for the choice of constructs is needed.

• The interpretation of the results in Figure 2H–I is challenging. For instance, at line 304, the authors state, "Interestingly, BTZ-induced genes are enriched in the 'overlap' category (Fig. 2I), suggesting that during proteasomal inhibition, genes typically activated by SKN-1C are under the transcriptional control of SKN-1A." However, it must be emphasized that the overlap represents a shared gene set between the SKN-1t-regulated genes and other stress response or mutant gene sets, rather than implying a direct causal relationship. An overlap between gene sets suggests a correlation, not causality, and it remains difficult to determine whether the observed phenomenon is due to direct control by SKN-1A or indirect effects through other mechanisms.

• In Figures 6B–C, the skn-1c mutant shows more resistance to AS compared to the skn-1ac mutant, suggesting that endogenous SKN-1A contributes to AS resistance in the absence of SKN-1C. Since SKN-1A is typically degraded by the proteasome, it is important to clarify whether AS treatment inhibits proteasome activity, thereby stabilizing SKN-1A and enabling its transcriptional function. While the transgenic experiments are understandable, the mechanism by which intact SKN-1A induces transcription without proteasome inhibition remains unclear and warrants further discussion.

• In Figures 7A–B, the inactivation of SKN-1A-dependent oxidative stress defenses in the png-1 null mutant is attributed to the lack of transcriptional activity of SKN-1A modified with N-glycans, rather than suggesting that PNG-1-mediated sequence editing actively promotes optimal oxidative stress defenses.

Minor

• The values stated in lines 194–196 do not match those shown in Figure 1B. The corrected statement should be: “The 938 genes upregulated by SKN-1t[NNNN] overlap much more with SKN-1t[NDNN] (749/938 = 80%) than with SKN-1t[DNDD] (232/938 = 25%). Although SKN-1t[DNDD] activates fewer genes overall (431), it has a relatively large proportion (125/431 = 29%) of uniquely upregulated genes not similarly increased by either SKN-1t[NNNN] or SKN-1t[NDNN].”

• In line 507, "Fig S9" should be corrected to "Fig S8."

Reviewer #2: This is a well-executed and insightful study that advances our understanding of the post-translational regulation and functional specialization of the SKN-1 isoforms in C. elegans, with clear relevance to the mammalian Nrf1/Nrf2 system. The authors elegantly demonstrate how N-glycosylation-dependent “sequence editing” fine-tunes the transcriptional output of SKN-1A (analogous to Nrf1), distinguishing its functions from SKN-1C (analogous to Nrf2). This work clarifies an important mechanistic question in the field and should be of broad interest to researchers studying stress responses, transcriptional regulation, proteostasis, and aging.

The manuscript is well written, the data are rigorous and thoughtfully interpreted, and the figures are clear and informative. I particularly appreciated the authors’ use of multiple lines of evidence (including engineered transgenes with site-specific Asn-to-Asp substitutions, isoform-specific mutants, and extensive transcriptomic profiling) to dissect the influence of sequence editing on gene regulation and organismal phenotypes. The logic of the experiments is compelling, and the findings are highly relevant to the field.

Major strengths:

The use of engineered SKN-1t constructs with graded sequence editing (Asn→Asp mutations) is an elegant way to interrogate the transcriptional consequences of SKN-1A activation.

The demonstration that sequence editing both enables unique SKN-1A targets (proteasome genes) and attenuates shared SKN-1C targets (oxidative stress genes) is novel and important.

The use of isoform-specific mutants is well executed and allows a clear functional dissection of SKN-1A versus SKN-1C roles across multiple stress paradigms.

The findings have direct implications for how cells interpret stress signals and mount isoform-specific transcriptional responses, which is highly relevant to therapeutic targeting of Nrf1/Nrf2 in disease.

Recommended Revisions:

Clarify the biological context of partial sequence editing: The intermediate phenotype of the SKN-1t[NDNN] construct is informative and well interpreted. A brief statement in the Discussion about the plausibility of partial editing under physiological conditions would help contextualize this construct’s relevance.

Improve figure legend clarity: The categories “high-D,” “overlap,” and “low-D” are used throughout the manuscript. While they are defined in the main text, summarizing these distinctions concisely in the figure legends would aid readability for those referencing figures independently.

Typos: A few minor typographical repetitions (e.g., “SKN-1t[DNDD] and SKN-1t[DNDD]”) appear in the Results section and should be corrected in the final version.

Strengthen translational relevance in Discussion: The parallels to mammalian Nrf1/Nrf2 are well drawn throughout the manuscript, but the Discussion could benefit from a slightly more direct reflection on potential implications for therapeutic targeting of Nrf1, especially in contexts like proteasome inhibitor therapeutics in cancer. This would underscore the translational relevance of the findings.

Reviewer #3: Major:

1. The authors did not provide enough introduction to the background of these findings. It’s really hard to follow the logic without some necessary information.

For example, why does exposure to BTZ activate SKN-1A? “wdr-23” and “xrep-4” were first mentioned in Figure 2H, but the function of “wdr-23” was introduced when describing Figure 3C.

2. Why do skn-1c mutants have a better survival rate at 4 mM PQ but not at 2 or 3 mM? Please confirm these results and provide a hypothesis.

3. The authors claimed that the four glycosylation sites are important for the regulation of downstream gene expression. It would be better if the authors could obtain the SKN-1 binding motif of each mutant and compare it with the existing SKN-1 motifs, such as: GSE246466; JASPAR; CIS-BP. These findings might help in understanding the mechanism of the transcriptional roles of these glycosylation sites.

Minor:

1. “Wild type” in the figures should not be italicized.

2. Line 694: Add version information to EdgeR.

3. Line 696: (Methods, RNAseq Analysis): “Genes with low expression were excluded using edgeR function filterByExpr...” → “...excluded using the edgeR function filterByExpr...”.

4. Line 711: “SKN-1t[NNNN] AND SKN-1t[NDNN] AND SKN-1t[DNDD]” → “SKN-1t[NNNN], SKN-1t[NDNN], and SKN-1t[DNDD]”.

5. Line 723 (Typo): “CRSIPR/Cas9 genome editing...” → “CRISPR/Cas9 genome editing...”.

6. Line 820: Add more details about the embryonic lethality. I think the worms used here are Day 1 adult animals. Lots of eggs from Day 5 adults might not hatch.

**Have all data underlying the figures and results presented in the manuscript been provided?**

Reviewer #1: Yes

Reviewer #2: **No: ** GEO accession GSE279082. This is mentioned in the manuscript information sheets in the beginning, but not present in the actual manuscript.

Reviewer #3: Yes

PLOS authors have the option to publish the peer review history of their article (what does this mean? ). If published, this will include your full peer review and any attached files.

**Do you want your identity to be public for this peer review?** For information about this choice, including consent withdrawal, please see our Privacy Policy .

Reviewer #1: No

Reviewer #2: **Yes: ** Senthil Radhakrishnan

Reviewer #3: **Yes: ** Lei Wang

**Figure resubmission:**
---

## [Decision Letter · Decision Letter 1]

PGENETICS-D-25-00366R1

Protein sequence editing defines distinct and overlapping functions of SKN-1A/Nrf1 and SKN-1C/Nrf2.

PLOS Genetics

Dear Dr. Lehrbach,

Thank you for submitting your revised manuscript to PLOS Genetics. After reviewing the updated version, we invite you to submit another revision that softens the conclusions presented in the text, in order to address the remaining concern raised by Reviewer 1.

Please submit your revised manuscript within 30 days Jul 15 2025 11:59PM. If you will need more time than this to complete your revisions, please reply to this message or contact the journal office at plosgenetics@plos.org. Please include the following items when submitting your revised manuscript:

We look forward to receiving your revised manuscript.

Kind regards,

Laura Bianchi

Academic Editor

PLOS Genetics

Marnie Blewitt

Section Editor

PLOS Genetics

Aimée Dudley

Editor-in-Chief

PLOS Genetics

Anne Goriely

Editor-in-Chief

PLOS Genetics

**Journal Requirements:**

1) When completing the data availability statement of the submission form, you indicated that you will make your data available on acceptance. We strongly recommend all authors decide on a data sharing plan before acceptance, as the process can be lengthy and hold up publication timelines. Please note that, though access restrictions are acceptable now, your entire data will need to be made freely accessible if your manuscript is accepted for publication. This policy applies to all data except where public deposition would breach compliance with the protocol approved by your research ethics board. If you are unable to adhere to our open data policy, please kindly revise your statement to explain your reasoning and we will seek the editor's input on an exemption. Please be assured that, once you have provided your new statement, the assessment of your exemption will not hold up the peer review process.

2) Please include the affiliation of Briar Jochim and Irini Topalidou in the online submission form. Please ensure that the affiliations of the authors listed on the manuscript title page do exactly match with the affiliations provided in the online submission form. NOTE: Affiliations should include a department (if applicable), an institution, a city, and a country.

3) Please ensure that the funders and grant numbers match between the Financial Disclosure field and the Funding Information tab in your submission form. Note that the funders must be provided in the same order in both places as well.  

**Reviewers' comments:**

Reviewer's Responses to Questions

**Comments to the Authors:**

Reviewer #1: The authors have significantly improved the manuscript and have addressed most of the reviewers’ comments.

However, there remains one critical point that is difficult to reconcile. Specifically, the authors claim that the inactivation of SKN-1A–dependent oxidative stress defenses in the png-1 null mutant is not due to the lack of transcriptional activity of SKN-1A modified with N-glycans, but rather due to the lack of sequence editing.

Based on the results shown in Figure 7A–B, it is unclear how this conclusion can be drawn. While it is possible that the inactivation results from the unedited form of SKN-1A, the data do not appear to exclude the alternative possibility that the N-glycosylation of SKN-1A itself compromises its transcriptional activity. Therefore, the authors’ conclusion regarding the irrelevance of N-glycosylation in this context may not be fully supported by the presented results.

Reviewer #2: The authors have satisfactorily addressed my comments

Reviewer #3: I have no further comments. The manuscript has significantly improved.

**Have all data underlying the figures and results presented in the manuscript been provided?**

Reviewer #1: Yes

Reviewer #2: Yes

Reviewer #3: Yes

PLOS authors have the option to publish the peer review history of their article (what does this mean? ). If published, this will include your full peer review and any attached files.

**Do you want your identity to be public for this peer review?** For information about this choice, including consent withdrawal, please see our Privacy Policy .

Reviewer #1: **Yes: ** Yukiko Yoshida

Reviewer #2: **Yes: ** Senthil Radhakrishnan

Reviewer #3: **Yes: ** Lei Wang

**Figure resubmission:**
---

## [Editor Report · Decision Letter 2]

Dear Dr Lehrbach,

We are pleased to inform you that your manuscript entitled "Protein sequence editing defines distinct and overlapping functions of SKN-1A/Nrf1 and SKN-1C/Nrf2." has been editorially accepted for publication in PLOS Genetics. Congratulations!

Yours sincerely,

Laura Bianchi

Academic Editor

PLOS Genetics

Marnie Blewitt

Section Editor

PLOS Genetics

Aimée Dudley

Editor-in-Chief

PLOS Genetics

Anne Goriely

Editor-in-Chief

PLOS Genetics

Comments from the reviewers (if applicable):

**Data Deposition**

http://datadryad.org/submit?journalID=pgenetics&manu=PGENETICS-D-25-00366R2

**Press Queries**

---

## [Editor Report · Acceptance letter]

PGENETICS-D-25-00366R2

Protein sequence editing defines distinct and overlapping functions of SKN-1A/Nrf1 and SKN-1C/Nrf2.

Dear Dr Lehrbach,

We are pleased to inform you that your manuscript entitled "Protein sequence editing defines distinct and overlapping functions of SKN-1A/Nrf1 and SKN-1C/Nrf2." has been formally accepted for publication in PLOS Genetics! Your manuscript is now with our production department and you will be notified of the publication date in due course.

With kind regards,

Anita Estes

PLOS Genetics

On behalf of:
